# Identify Critical KV Cache in LLM Inference from an Output Perturbation Perspective

## Abstract

Large language models have driven numerous paradigm shifts in the field of natural language processing, achieving remarkable success in various real-world applications through scaling model size and leveraging long-sequence context reasoning. However, the transformer architecture, which relies on self-attention, incurs substantial storage and runtime costs when handling long-sequence inference, particularly due to the generation of extensive Key-Value (KV) cache. Recent studies aim to mitigate storage and latency issues while maintaining output quality by reducing the KV cache size, through the elimination of less critical entries, yet they rely on a basic empirical intuition of identifying critical cache entries based solely on top attention weights. In this paper, we present the first formal investigation into the problem of identifying critical KV cache entries from the perspective of attention output perturbation. By analyzing the output perturbation caused when only critical KV cache entries are used instead of the entire cache, we reveal that, in addition to the commonly used attention weights, the value states within KV entries and the pretrained parameters matrix are also important. Based on this finding, we propose a novel perturbation-constrained selection algorithm to identify critical cache entries by optimizing the worst-case output perturbation. Extensive evaluations on 16 datasets from Longbench, along with detailed empirical analysis, have comprehensively confirmed the effectiveness of constraining output perturbation perspective in identifying critical KV cache. When combined with state-of-the-art cache eviction methods, it can achieve up to an additional 34% cache memory savings while maintaining the same generation quality.

## 1 Introduction

Autoregressive large language models (LLMs) based on the transformer architecture have achieved remarkable success, being widely applied in various downstream tasks such as dialogue systems (Yi et al., 2024), chatbots (Achiam et al., 2023), intelligent agents (Wang et al., 2024), and code generation (Gu, 2023). However, the quadratic computational cost inherent in the transformer's self-attention mechanism poses significant challenges for practical deployment. To mitigate this, LLMs often use a Key-Value (KV) cache, which stores intermediate results from the self-attention mechanism. Each KV cache entry corresponds to the KV states of a past token, thus allowing for the bypassing of recomputation of these tokens during autoregressive generation. However, as sequence lengths increase, the number of the KV cache entries expands correspondingly. This expansion in KV cache not only leads to considerable GPU memory overhead but also significantly increases IO latency, hindering the deployment in real-world applications (Sun et al., 2024a).

Recent researches identify that only a subset of KV cache entries substantially contribute to the output of the self-attention mechanism (Zhang et al., 2024b; Liu et al., 2024a; Tang et al., 2024a). Therefore, many methods known as *cache eviction* have been developed to reduce the KV cache size to fit within a given GPU memory budget by evicting non-critical entries during long-sequence inference. These budget-constrained methods effectively save GPU memory and improve subsequent decoding speed. Initial findings show that recent KV cache entries are more critical for future token generation, prompting the development of techniques that prioritize retaining KV entries within a recent window (Beltagy et al., 2020; Xiao et al., 2023). However, this approach risks losing essential information from longer sequences. To address this, H2O (Zhang et al., 2024b) and Scissorhands (Liu et al., 2024a) observe the power-law distribution of attention weights: a small fraction of KV

cache entries consistently dominates the majority of attention weights, which aligns closely with the concept of cache entry criticality during inference. These methods introduce frameworks that leverage accumulated attention weights to identify and preserve critical cache entries. Following the observation, a series of subsequent works (Adnan et al., 2024; Li et al., 2024; Feng et al., 2024) have further improved performance by refining the attention weight accumulation mechanism or incorporating additional operations like polling to better retain key information. Although the term "*critical cache entry*" remains an abstract concept without formal definition, current approaches often assume that cache entries with higher attention weights — determined by the similarity between key states in the KV cache and the target query state — indicate the critical cache entries. This assumption raises two key questions:

1. *What criteria determine the critical KV cache?*
2. *Is reliance solely on attention weights sufficient for identifying critical cache entries?*

In this paper, we formally define the problem of critical cache identification from the perspective of output perturbation, introducing a theoretical framework that optimizes the perturbation by bounding it within worst-case scenarios. Specifically, we formalize the problem as minimizing the output perturbation while substituting the entire KV cache with only the critical cache entries. To quantify this perturbation, we employ the $L_1$ distance and derive its upper bound, corresponding to the worst-case perturbation. Our analysis shows that this upper bound is influenced by both the attention weights and the value states projected through the parameter matrix. Building on this foundation, we propose a perturbation-constrained selection algorithm that goes beyond mere reliance on attention weights, underscoring the significance of previously overlooked value states and pretrained parameter matrix in identifying critical cache entries.

Building on our theoretical framework for critical cache identification based on output perturbation, we replace previous strategies that rely solely on attention weights by seamlessly integrating our algorithm into state-of-the-art (SOTA) cache eviction methods. Using 16 datasets from LongBench, we conduct extensive evaluations across various budgets, demonstrating that our algorithm selects critical cache entries more accurately, effectively improving the post-eviction generation quality. We conduct further empirical analysis to evaluate the benefits of our algorithm, leading to two key conclusions: (1) across different methods, contexts, budgets, and models, our algorithm consistently reduces output perturbation in most attention heads, and (2) its advantages accumulate across layers, significantly lowering final-layer output perturbation. These findings show that our algorithm consistently improves post-eviction generation quality and confirms the effectiveness in identifying critical cache entries. Our contributions can be summarized as follows:

1. We point that current cache eviction methods neglect the crucial problem of KV cache identification. To address this, we propose using output perturbation as criteria to determine critical KV cache entries. Our analysis shows that attention weights alone are insufficient, as the value states projected by the parameter matrix also play a significant role.

2. Building on constraining the worst-case output perturbation, we propose a novel critical entry selection algorithm. When integrated into SOTA eviction methods, comprehensive evaluations across 16 datasets demonstrate it consistently improves the generation quality.

3. Further empirical analysis examines and confirms the benefits of our perturbation-constrained selection algorithm, which also highlights the significant potential for optimizing critical cache selection from the theoretical perspective of output perturbation.

## 2 RELATED WORKS

The inference cost of LLMs is primarily determined by the size of model parameters and KV cache. Early strategies like parameter quantization and network pruning successfully reduce model parameters. However, with the rise of applications such as long-text summarization (Laban et al., 2023), multi-turn QA (Yang et al., 2018), and techniques like Chain of Thought (Wei et al., 2022), the increasing sequence length has led to substantial growth in KV cache size. Consequently, recent research has shifted focus toward reducing KV cache size to enable efficient long-sequence inference, primarily through two orthogonal approaches: *KV cache quantization* and *KV cache eviction*.

**KV cache quantization** refers to the application of quantization techniques to reduce the size of the KV cache by lowering the precision of individual cache entries (Liu et al., 2024b; Hooper et al.,

2024). For example, this can involve quantizing the original 16-bit KV cache entries to 2-bit or 4-bit precision. However, such approaches typically retain all KV cache entries (Hooper et al., 2024). As a result, in autoregressive inference with long sequences, they cannot effectively compress the KV cache to fit within a specified budget. These methods are fundamentally orthogonal to the cache eviction methods, and could further enhance performance by incorporation.

**KV cache eviction** focuses on retaining only critical KV cache entries while evicting non-critical ones to reduce cache size. Early methods (Xiao et al., 2023), which preserved recent entries, risked losing important information in long sequences. Techniques like H2O (Zhang et al., 2024b) and Scissorhands (Liu et al., 2024a) used accumulated attention scores to identify key entries, aiming to retain crucial context. Subsequent works refined these methods (Ge et al., 2024b; Adnan et al., 2024; Ge et al., 2024a; Li et al., 2024), with SnapKV (Li et al., 2024) achieving the SOTA performance through introducing window-based attention weight accumulation and pooling operations. More recent approaches, such as Pyramid (Yang et al., 2024; Zhang et al., 2024a) and AdaKV (Feng et al., 2024), improve post-eviction generation by allocating budgets based on the characteristics of attention heads. However, these methods are largely empirical, relying heavily on attention weights to identify critical entries. Our paper introduces a novel perturbation-constrained selection algorithm based on in-depth analysis from an output perturbation perspective. This algorithm seamlessly integrates into existing cache eviction methods without altering underlying accumulation processes. In our work, we demonstrate its effectiveness by applying it to SnapeKV, Pyramid, and AdaKV, all showing consistent improvements in post-eviction generation quality under varying budgets.

## 3 CRITICAL KV CACHE ENTRY SELECTION

We commence with a commonly held observation (Zhang et al., 2024b; Li et al., 2024) that a subset of critical KV cache entries can adequately represent the entire KV cache during the computation of self-attention mechanism, producing an output that is a close approximation, if not identical. The preliminaries about the relationship between KV cache and generation output are introduced in Section 3.1. Based on that, we formalize the problem of identifying critical cache entries from the perspective of output perturbation (Definition 1) in Section 3.2. Subsequently, in Section 3.3, we formalize the output perturbation and derive its upper bound. Then, we propose a two-stage greedy algorithm in Section 3.4 that constrains worst-case perturbations for selecting critical entries, with theoretical analysis provided in Section 3.5. Finally, in Section 3.6 we integrate the algorithm into current SOTA cache eviction methods.

### 3.1 PRELIMINARIES

LLMs utilizing the multi-head self-attention mechanism operate with an autoregressive generation approach. In this setup, each decoding step leverages the most recently generated token to predict the next one. To illustrate this process, we focus on a single attention head as an example. Let $X \in \mathbb{R}^{n \times d}$ denote the embedding matrix for all tokens in the sequence, with $x = X_{-1,:} \in \mathbb{R}^{1 \times d}$ representing the embedding vector of the most recent token, which serves as input at the current time step. The parameter matrices, denoted by $W^Q$, $W^K$, and $W^V \in \mathbb{R}^{d \times d_h}$ are used to map the token embeddings into their respective Query, Key, and Value states with head dimension $d_h$ as follows:

$$q = xW^Q; K = XW^K; V = XW^V \tag{1}$$

During the decoding phase, the Key and Value states of previously generated tokens (represented by $X$) are stored in the KV cache, allowing for the elimination of redundant computation. Accordingly, the query $q$, derived from the most recent token $x$, attends to the cached Key $K$ to compute the attention weights $A$. These weights are then applied to the cached Value $V$, producing an intermediate output. This intermediate result is subsequently transformed into the final output $o$ of the self-attention mechanism by the output parameter matrix $W^O \in \mathbb{R}^{d_h \times d}$:

$$o = AVW^O, \text{ where } A = \text{softmax}\left(qK^T/\sqrt{d}\right) \tag{2}$$

### 3.2 WHAT CRITERIA DETERMINE THE CRITICAL KV CACHE?

Recent research has demonstrated only a small portion of critical KV cache entries do substantially contribute to the attention output (Zhang et al., 2024b; Liu et al., 2024a). This insight presents

promising opportunities to reduce inference costs by evicting a large number of non-critical KV cache entries Li et al. (2024); Zhang et al. (2024a); Feng et al. (2024); Ge et al. (2024b); Adnan et al. (2024); Ge et al. (2024a).However, the key challenge lies in accurately identifying the critical KV cache entries. Ideally, from a high-level perspective, the set of critical KV cache entries should completely represent the entire cache, ensuring for given query state, the selected entries yield the same attention output as the full set of KV pairs. In practice, the number of selected critical cache entries will be constrained by a predefined budget, which is closely tied to the computational resources available in downstream deployments. Consequently, our goal shifts toward minimizing the output perturbation introduced by the replacement. So, the problem can be reformulated as follows.

**Definition 1** (Critical KV Cache Identification Problem)**.** *Given a critical cache budget $b$, the task is to select $b$ critical KV cache entries $\langle \hat{K}, \hat{V} \rangle$ from a total of $n$ cache entries $\langle K, V \rangle$, with the goal of minimizing the perturbation in the attention output $o$. By using the $L_1$ distance $\mathcal{L}$ for quantification, the objective is formalized as:*

$$\underset{selection \ of \ \langle \hat{K}, \hat{V} \rangle}{\arg\min} \quad \mathcal{L} = \|o - \hat{o}\|_1 \tag{3}$$

*where $\hat{o}$ represents the attention output produced by the selected $\langle \hat{K}, \hat{V} \rangle$.*

### 3.3 ARE ATTENTION WEIGHTS SUFFICIENT FOR IDENTIFYING CRITICAL CACHE ENTRIES?

According to Definition 1, the goal of identifying critical KV cache entries is to minimize the perturbation $\mathcal{L} = \|o - \hat{o}\|_1$. To achieve this, we can employ an additive masking $\mathcal{M}$ to simulate the removal of non-critical cache entries' contributions to the final output $\hat{o}$, thereby altering $\hat{o}$.

$$\hat{o} = A'VW^O, \ A' = \text{softmax}\left(\mathcal{M} + qK^T/\sqrt{d}\right), \text{where } \mathcal{M}_i = \begin{cases} -\infty & \text{if } K_i \text{ and } V_i \text{ are non-critical} \\ 0 & \text{otherwise.} \end{cases} \tag{4}$$

Thus, the perturbation $\mathcal{L}$ can be further expressed as:

$$\mathcal{L} = \|(A - A')VW^O\|_1 \tag{5}$$

**Theorem 1.** *By introducing a mask $\mathcal{N} \in \mathbb{R}^n$ applied through element-wise multiplication denoted by $\odot$, we can establish the relation between $A'$ and $A$ as follows:*

$$A' = \frac{\mathcal{N} \odot A}{\sum_{i=1}^n \mathcal{N}_i A_i} \quad \text{where } \mathcal{N}_i = \begin{cases} 0 & \text{if } K_i, V_i \text{ is non-critical} \\ 1 & \text{otherwise.} \end{cases} \text{ and } \sum_{i=1}^n \mathcal{N}_i = b \tag{6}$$

*Proof.* Let $a = qK^T/\sqrt{d}$, we can express the attention weights $A'$ under critical cache entries as:

$$A' = \frac{exp(\mathcal{M} + a)}{\sum_{i=1}^n exp(\mathcal{M} + a)_i} = \frac{\mathcal{N} \odot exp(a)}{\sum_{i=1}^n \mathcal{N}_i exp(a)_i} = \mathcal{N} \odot \frac{exp(a)}{\sum_{i=1}^n exp(a)_i} \frac{\sum_{i=1}^n exp(a)_i}{\sum_{i=1}^n \mathcal{N}_i exp(a)_i} \tag{7}$$

Considering $A = \frac{exp(a)}{\sum_{i=1}^n exp(a)_i}$, thus $\sum_{i=1}^n \mathcal{N}_i A_i = \frac{\sum_{i=1}^n \mathcal{N}_i exp(a)_i}{\sum_{i=1}^n exp(a)_i}$. Therefore, $A' = \frac{\mathcal{N} \odot A}{\sum_{i=1}^n \mathcal{N}_i A_i}$. $\square$

Theorem 1 utilizes a multiplicative mask $\mathcal{N}$ to quantifies how their selection impacts the attention weights. However, directly minimizing $\mathcal{L}$ for critical cache selection is challenging due to complex matrix operations it requires. Thus we turn to establish an upper bound $\theta$, as shown in Theorem 2.

**Theorem 2.** *The output perturbation $\mathcal{L}$ can be bounded by $\theta$:*

$$\mathcal{L} \leq \theta = C - \left(2 - \frac{1}{\sum_{i=1}^n \mathcal{N}_i A_i}\right) \sum_{i=1}^n \mathcal{N}_i A_i \|\mathcal{V}_{i,:}\|_1, \tag{8}$$

*where $C$ denotes the $\sum_{i=1}^n A_i \|\mathcal{V}_{i,:}\|_1$ and $\mathcal{V} \in \mathbb{R}^{n \times d} = VW^O$ denotes all projected values states through parameter matrix $W^O$.*

---

**Algorithm 1** Perturbation-Constrained Selection for Critical Cache Entry Identification

---

**Input**: Budgets $b$, Query State $q$, KV Cache Entries $K, V$, Parameter Matrix $W^O$, Hyper Parameter $\alpha = 0.25$
**Output**: Critical Cache Entries $\hat{K}, \hat{V}$

 1: initialize empty cache $\hat{K}, \hat{V}$
 2: $A = \text{softmax}(qK^T); \mathcal{V} = VW^O$
 3: $\mathcal{A} = A \odot (L_1 \text{ norm of each rows in } \mathcal{V})$
 4: $b' = b \times \alpha; b'' = b - b'$
 5: **for** $A_i, K_i, V_i \in A, K, V$ **do** $\qquad\qquad\qquad\qquad\qquad\qquad$ ▷ Start of **Stage 1**
 6: $\quad$ **if** $A_i \in \text{Top}_k(A, b')$ **then**
 7: $\qquad$ add $K_i, V_i$ to $\hat{K}, \hat{V}$
 8: $\qquad$ remove $\mathcal{A}_i, K_i, V_i$ from $\mathcal{A}, K, V$
 9: $\quad$ **end if**
10: **end for** $\qquad\qquad\qquad\qquad\qquad\qquad\qquad\qquad\qquad\qquad\qquad$ ▷ End of **Stage 1**
11: **for** $\mathcal{A}_i, K_i, V_i \in \mathcal{A}, K, V$ **do** $\qquad\qquad\qquad\qquad\qquad$ ▷ Start of **Stage 2**
12: $\quad$ **if** $\mathcal{A}_i \in \text{Top}_k(\mathcal{A}, b'')$ **then**
13: $\qquad$ add $K_i, V_i$ to $\hat{K}, \hat{V}$
14: $\quad$ **end if**
15: **end for** $\qquad\qquad\qquad\qquad\qquad\qquad\qquad\qquad\qquad\qquad\qquad$ ▷ End of **Stage 2**
16: **return** Critical Cache Entries $\hat{K}, \hat{V}$

---

*Proof.* Let $\mathcal{V} \in \mathbb{R}^{n \times d} = VW^O$ denote all projected value states, thus:

$$\mathcal{L} = \left\| \left( A - \frac{\mathcal{N} \odot A}{\sum_{i=1}^n \mathcal{N}_i A_i} \right) \mathcal{V} \right\|_1 = \left\| \sum_{i=1}^n \left( A_i - \frac{\mathcal{N}_i A_i}{\sum_{i=1}^n \mathcal{N}_i A_i} \right) \mathcal{V}_{i,:} \right\|_1 \qquad (9)$$

$$\leq \theta = \sum_{i=1}^n \left\| \left( A_i - \frac{\mathcal{N}_i A_i}{\sum_{i=1}^n \mathcal{N}_i A_i} \right) \mathcal{V}_{i,:} \right\|_1 = \sum_{i=1}^n \left| A_i - \frac{\mathcal{N}_i A_i}{\sum_{i=1}^n \mathcal{N}_i A_i} \right| \times \| \mathcal{V}_{i,:} \|_1 \qquad (10)$$

Given that the multiplicative mask $\mathcal{N}$ is either 0 or 1, the index set $i \in [1, n]$ can be split into $I_0$ and $I_1$, according to its value. Thus:

$$\mathcal{L} \leq \theta = \sum_{i \in I_0} A_i \| \mathcal{V}_{i,:} \|_1 + \sum_{i \in I_1} \left( \frac{A_i}{\sum_{i=1}^n \mathcal{N}_i A_i} - A_i \right) \| \mathcal{V}_{i,:} \|_1 \qquad (11)$$

Let $C$ represent $\sum_{i=1}^n A_i \| \mathcal{V}_{i,:} \|_1$, a constant independent of the selection of critical entries. We can express $\sum_{i \in I_0} A_i \| \mathcal{V}_{i,:} \|_1$ as $C - \sum_{i \in I_1} A_i \| \mathcal{V}_{i,:} \|_1$. Thus:

$$\mathcal{L} \leq \theta = C - \sum_{i \in I_1} A_i \| \mathcal{V}_{i,:} \|_1 + \sum_{i \in I_1} \left( \frac{A_i}{\sum_{i=1}^n \mathcal{N}_i A_i} - A_i \right) \| \mathcal{V}_{i,:} \|_1 \qquad (12)$$

$$= C + \sum_{i \in I_1} \left( \frac{A_i}{\sum_{i=1}^n \mathcal{N}_i A_i} - 2A_i \right) \| \mathcal{V}_{i,:} \|_1 = C - \left( 2 - \frac{1}{\sum_{i=1}^n \mathcal{N}_i A_i} \right) \sum_{i=1}^n \mathcal{N}_i A_i \| \mathcal{V}_{i,:} \|_1 \qquad (13)$$

$$\square$$

We can observe that $\theta$ encompasses not only the attention weights but also the projected value states. This highlights that prior selection methods relying solely on attention weights are suboptimal.

### 3.4 IDENTIFY CRITICAL CACHE ENTRIES BY CONSTRAINING WORST-CASE PERTURBATION.

Drawing on optimization strategies in machine learning, we propose lowering the upper bound of perturbation, effectively constraining the worst-case perturbation and thereby reducing actual perturbations for identifying critical cache entries. However, directly minimizing the upper bound $\theta$ remains non-trivial. To balance both the complexity and selection effectiveness, we introduce a two-stage greedy perturbation-constrained selection Algorithm 1, specifically designed to lower the perturbation upper bound for critical cache entry identification.

In this algorithm, the total budget $b$ is divided into two portions based on a hyperparameter $\alpha$. In the first stage, a small fraction of the budget, $b' = b \times \alpha$, is allocated to prioritize KV cache entries with high attention weights. In the second stage, the remaining budget, $b'' = b - b'$, is used to consider both the norms of the projected value states and the attention weights. This two-stage selection employs a Top-K operation to effectively constrain the worst-case perturbation. To substantiate the effectiveness of our proposed algorithm, we provide a theoretical analysis in the following section.

## 3.5 THEORETICAL ANALYSIS OF PERTURBATION-CONSTRAINED SELECTION ALGORITHM

Our proposed algorithm comprises two stages, referred to as stage 1 and stage 2, with the latter serving as the algorithm's core component. Under the guarantee provided by Assumption 1, the selection in Stage 1 ensures that Stage 2 adheres to the constraints on perturbations, as formalized in Theorem 3. Let $\mathcal{N}'$ and $\mathcal{N}''$ represent the selections from the stage 1 and 2, respectively, satisfying: $\sum_{i=1}^{n} \mathcal{N}'_i = b'$ and $\sum_{i=1}^{n} \mathcal{N}''_i = b''$. Thus, the overall selection is $\mathcal{N} = \mathcal{N}' + \mathcal{N}''$.

**Assumption 1.** *In the first stage, the small portion of the overall budget $b' = b \times \alpha$ is sufficient to collect the cache entries corresponding to the highest attention weights, ensuring their cumulative attention weights $\sigma$ exceed half of the total, i.e., $\sigma = \sum_{i=1}^{n} \mathcal{N}'_i A_i = \sum Top_k(A, b') > 0.5$.*

We can simply set $\alpha$ to a small fraction, such as 0.25, to ensure that Assumption 1 holds. This is primarily determined by two key factors. Firstly, the inherent power-law distribution of attention weights (Zhang et al., 2024b), where a small subset of cache entries accounts for the majority of the attention weights. Secondly, in practical compression scenarios, the overall budget is generally maintained at a reasonable level to avoid catastrophic degradation in generation quality after eviction. Therefore, Assumption 1 is universally satisfied across most attention heads during actual operations. Experimental analysis further validate the robustness of Assumption 1 in Appendix A and the effectiveness of stage 1 in Appendix D.

**Theorem 3.** *Given the stage 1 selection $\mathcal{N}'_i$, the objective $\mathcal{N}''_i$ of stage 2 is to minimize an upper bound $\hat{\theta}$ of the output perturbation $\mathcal{L}$, using the remaining budget $b'' = b - b'$.*

$$\arg\min_{\mathcal{N}''_i} \hat{\theta} \quad where \quad \hat{\theta} = C' - \left(2 - \frac{1}{\sigma}\right) \sum_{i=1}^{n} \mathcal{N}''_i A_i \|\mathcal{V}_{i,:}\|_1 \tag{14}$$

$$subject\ to \sum_{i=1}^{n} \mathcal{N}''_i = b'', \quad C' = C - \left(2 - \frac{1}{\sigma}\right) \sum_{i=1}^{n} \mathcal{N}'_i A_i \|\mathcal{V}_{i,:}\|_1. \tag{15}$$

*Proof.* From Assumption 1, the first stage selection ensures: $\sum_{i=1}^{n} \mathcal{N}_i A_i > \sum_{i=1}^{n} \mathcal{N}'_i A_i = \sigma > 0.5$, leading to the inequality: $2 - \frac{1}{\sum_{i=1}^{n} \mathcal{N}_i A_i} > 2 - \frac{1}{\sigma} > 0$.

$$\theta = C - \left(2 - \frac{1}{\sum_{i=1}^{n} \mathcal{N}_i A_i}\right) \sum_{i=1}^{n} (\mathcal{N}'_i + \mathcal{N}''_i) A_i \|\mathcal{V}_{i,:}\|_1 \tag{16}$$

$$< C - \left(2 - \frac{1}{\sigma}\right) \sum_{i=1}^{n} \mathcal{N}'_i A_i \|\mathcal{V}_{i,:}\|_1 - \left(2 - \frac{1}{\sigma}\right) \sum_{i=1}^{n} \mathcal{N}''_i A_i \|\mathcal{V}_{i,:}\|_1 \tag{17}$$

Let $C' = C - \left(2 - \frac{1}{\sigma}\right) \sum_{i=1}^{n} \mathcal{N}'_i A_i \|\mathcal{V}_{i,:}\|_1$, then we can derive a new upper bound $\hat{\theta}$ for $\mathcal{L}$ factoring by second stage selection $\mathcal{N}''_i$: $\theta < C' - \left(2 - \frac{1}{\sigma}\right) \sum_{i=1}^{n} \mathcal{N}''_i A_i \|\mathcal{V}_{i,:}\|_1 = \hat{\theta}$ Thus, minimizing $\hat{\theta}$ corresponds to selecting the $b''$ entries with the highest values of $\mathcal{A}_i = A_i \|\mathcal{V}i,:\|_1$, as implemented in the stage 2 selection (Algorithm 1). $\square$

Theorem 3 demonstrates that our second stage selection directly minimizes an upper bound of output perturbation for identifying critical cache entries. Unlike traditional strategies that rely solely on high attention weights for entry selection, the second stage of our algorithm jointly leverages both the attention weights and the value states projected through the parameter matrix, to directly constrain the worst-case output perturbation.

## 3.6 INTEGRATING INTO SOTA CACHE EVICTION METHODS

We demonstrate the superiority of our proposed algorithm in application by integrating into the existing cache eviction methods which solely rely on accumulated attention weights for critical entries selection. Current SOTA cache eviction workflow is established by SnapKV, which introducing an observation window mechanism to stably accumulate attention weights and further employ the max pooling operations to prevent the omission of key information. Building on this, subsequent researches note the uneven distribution of critical cache entries across different heads, leading to the development of budget allocation strategies across heads for further optimization. Method, like Pyramid, employs a pyramid-like patterns for budget allocations for heads across different layers, while its pre-fixed encountering adaptability challenges with various LLMs. The latest innovation,

---

**Algorithm 2** Observation Window Based Cache Eviction Workflow.

---

**Input**: All Query States $Q \in \mathbb{R}^{n \times d_h}$, KV Cache Entries $K, V \in \mathbb{R}^{n \times d_h}$, Window Size $n'$
**Output**: Critical Cache Entries $\hat{K}, \hat{V}$

1: allocating budget $b$ for one head // refined by Pyramid and AdaSnap based on Snap
2: $\hat{Q} = Q[-n' :, :]$ // extract query states in observation window
3: $A = \text{softmax}(\hat{Q}K^T)$ ; $\bar{A} = A.\text{mean}(dim = 0)$ // calculate attention weights
4: $\bar{A}' = \text{maxpooling}(\bar{A})$ // max pooling across cache entries
5: **if** using regular selection **then**
6:     select $b$ critical cache entries $\hat{K}, \hat{V}$ according to $\text{Top}_k(\bar{A}', b)$
7: **else if** using our selection **then**
8:     select $b' = b \times \alpha$ critical entries $\hat{K}, \hat{V}$ according to $\text{Top}_k(\bar{A}', b')$
9:     $\mathcal{V} = VW^O$; $\mathcal{A} = \bar{A} \odot (L_1$ norm of each rows in $\mathcal{V})$
10:    append remaining $b'' = b \times (1 - \alpha)$ critical entries $\hat{K}, \hat{V}$ according to $\text{Top}_k(\mathcal{A}, b'')$
11: **end if**
12: **return** $\hat{K}, \hat{V}$

---

Ada-KV, dynamically detects the varying numbers of critical KV cache entries among different attention heads during runtime. This allows for flexible budget scheduling between heads, thereby significantly enhancing the quality of output generation.

Overall, this series of observation window-based cache eviction methods can be systematically unified as Algorithm 2. The entire cache eviction workflow can be divided into two main parts: the first is its budget allocation strategy across heads (line 1), and the second (lines 2—7) is the observation window mechanism for attention weights accumulation. Our algorithm integrates seamlessly with existing eviction methods without significant modifications, requiring only the additional computation of projected value states in line10.

## 4 APPLICATION IN CACHE EVICTION METHODS

### 4.1 DATASETS

For a comprehensive evaluation, we adopt "LongBench," a benchmark widely used for long-sequence tasks assessment (Li et al., 2024; Zhang et al., 2024a; Feng et al., 2024). It consists of 16 datasets across 6 task domains: single-document question answering (QA) (Kočiský et al., 2018; Dasigi et al., 2021), multi-document QA (Yang et al., 2018; Ho et al., 2020; Trivedi et al., 2022), summarization (Huang et al., 2021; Zhong et al., 2021; Fabbri et al., 2019), few-shot learning (Joshi et al., 2017; Gliwa et al., 2019; Li & Roth, 2002), synthetic tasks (Bai et al., 2023), and code generation (Guo et al., 2023; Liu et al., 2023). The overall average token length across all 16 datasets is 6,711. These datasets encompass a wide range of input sequence lengths, with average token counts ranging from 1,235 to 18,409. This variation places significant demands on KV cache memory, making them ideal for assessing KV cache eviction methods under various memory budgets $b \in \{128, 256, 512, 1024\}$. For each dataset, we apply the LongBench-recommended metrics, which assign a maximum quality score of 100. We then report the average scores for each task domain. Detailed information for each dataset can be found in Appendix C.

### 4.2 SETTINGS

We select two advanced open-source LLMs for evaluation: Mistral-7B-Instruct-v0.3 (Mistral-7B) (Jiang et al., 2023) and Llama-3.1-8B-Instruct (Llama-3.1-8B) (Dubey et al., 2024), which support maximum sequence lengths of 32K and 128K, respectively. We integrate our algorithm with three SOTA cache eviction methods—SnapKV (Li et al., 2024), Pyramid (Zhang et al., 2024a), and AdaKV (Feng et al., 2024)—and compared the generated output quality before and after integration. These cache eviction methods are applied directly to both models, without any additional training. All experiments are conducted on a single A800-80G GPU, and greedy decoding is employed during generation to ensure result stability. The hyper-parameter $\alpha$ in Algorithm 1 was set to 0.25 for all experiments. Other fundamental settings for SnapKV, Pyramid and AdaKV are kept as originally defined, without any modifications when integrating with our algorithm. For instance, the kernel size for max-pooling remained at 7, and the observation window size was set to 32.

Table 1: Integration into SnapKV

| SnapKV | | 6711 Full Cache | b = 128 | | b = 256 | | b = 512 | | b = 1024 | |
|---|---|---|---|---|---|---|---|---|---|---|
| | | | w/o ours | w/ ours | w/o ours | w/ ours | w/o ours | w/ ours | w/o ours | w/ ours |
| Llama-3.1-8B | Single-Doc. QA | 43.80 | 34.44 | **35.56** | 37.62 | **38.34** | 40.08 | **41.49** | 42.70 | **42.75** |
| | Multi-Doc. QA | 44.08 | 41.36 | **41.80** | 42.36 | **42.79** | 43.16 | **43.51** | **43.92** | 43.87 |
| | Summarization | 29.23 | 21.77 | **22.18** | 23.33 | **24.02** | 24.91 | **25.35** | **26.48** | 26.47 |
| | Few-shot | 69.24 | 59.61 | **60.00** | 63.42 | **63.91** | 67.22 | **66.99** | 67.81 | **68.51** |
| | Synthetic | 54.46 | 52.24 | **52.48** | 54.23 | **54.51** | 54.49 | **54.49** | **54.16** | 54.14 |
| | Code | 59.66 | 53.57 | **54.22** | 56.38 | **57.20** | 58.42 | **58.75** | 59.08 | **59.45** |
| | Ave. | 49.20 | 42.70 | **43.25** | 45.09 | **45.66** | 47.00 | **47.41** | 48.08 | **48.25** |
| Mistral-7B | Single-Doc. QA | 41.12 | 33.37 | **33.92** | 37.68 | **38.17** | 39.27 | **40.42** | **39.70** | 39.63 |
| | Multi-Doc. QA | 39.13 | **36.72** | 36.66 | **37.50** | 37.34 | 38.21 | **38.36** | 39.21 | **39.40** |
| | Summarization | 29.35 | 21.54 | **21.85** | **23.30** | 23.24 | 24.41 | **24.71** | 26.07 | **26.18** |
| | Few-shot | 70.57 | 59.03 | **60.53** | 65.48 | **66.02** | 68.26 | **68.44** | 69.41 | **69.97** |
| | Synthetic | 51.50 | 49.25 | **49.75** | 50.50 | **50.75** | 52.25 | **52.25** | 52.00 | **52.00** |
| | Code | 59.98 | 53.70 | **54.35** | **57.68** | 57.39 | 59.30 | **59.54** | **60.10** | 59.93 |
| | Ave. | 47.72 | 41.12 | **41.69** | 44.27 | **44.41** | 45.85 | **46.21** | 46.71 | **46.84** |

Table 2: Integration into Pyramid

| Pyramid | | 6711 Full Cache | b = 128 | | b = 256 | | b = 512 | | b = 1024 | |
|---|---|---|---|---|---|---|---|---|---|---|
| | | | w/o ours | w/ ours | w/o ours | w/ ours | w/o ours | w/ ours | w/o ours | w/ ours |
| Llama-3.1-8B | Single-Doc. QA | 43.80 | 34.85 | **35.09** | 37.56 | **38.05** | 40.09 | **40.44** | **42.31** | 42.25 |
| | Multi-Doc. QA | 44.08 | **41.36** | **41.36** | 42.33 | **42.53** | 43.10 | **43.37** | 43.83 | **44.18** |
| | Summarization | 29.23 | 21.72 | **22.44** | 23.58 | **23.94** | 24.74 | **25.18** | 26.27 | **26.56** |
| | Few-shot | 69.24 | 59.85 | **60.29** | 63.62 | **64.19** | 66.47 | **67.00** | 67.89 | **68.00** |
| | Synthetic | 54.46 | 52.87 | **52.98** | 54.19 | **54.53** | **54.56** | 54.47 | 54.08 | **54.37** |
| | Code | 59.66 | 53.06 | **53.79** | 54.33 | **56.11** | 57.02 | **57.50** | 58.80 | **58.85** |
| | Ave. | 49.20 | 42.82 | **43.19** | 44.90 | **45.46** | 46.65 | **46.99** | 47.92 | **48.09** |
| Mistral-7B | Single-Doc. QA | 41.12 | **36.28** | 36.18 | **36.28** | 36.18 | 38.46 | **38.72** | 39.36 | **39.45** |
| | Multi-Doc. QA | 39.13 | 36.78 | **37.15** | 36.78 | **37.15** | 37.90 | **38.11** | 39.03 | **39.31** |
| | Summarization | 29.35 | 23.13 | **23.25** | 23.13 | **23.25** | 24.25 | **24.45** | 25.78 | 25.77 |
| | Few-shot | 70.57 | 65.77 | **66.86** | 65.77 | **66.86** | 68.70 | 69.18 | 69.73 | **70.00** |
| | Synthetic | 51.50 | 50.25 | **50.50** | 50.25 | **50.50** | 50.50 | **51.00** | 52.25 | 51.00 |
| | Code | 59.98 | 56.08 | **56.72** | 56.08 | **56.72** | 58.07 | **58.59** | 59.34 | **59.66** |
| | Ave. | 47.72 | 41.15 | **41.63** | 43.66 | **44.05** | 45.32 | **45.66** | 46.56 | 46.56 |

Table 3: Integration into AdaKV

| AdaKV | | 6711 Full Cache | b = 128 | | b = 256 | | b = 512 | | b = 1024 | |
|---|---|---|---|---|---|---|---|---|---|---|
| | | | w/o ours | w/ ours | w/o ours | w/ ours | w/o ours | w/ ours | w/o ours | w/ ours |
| Llama-3.1-8B | Single-Doc. QA | 43.80 | 35.42 | **35.92** | 38.59 | **38.68** | 40.45 | **40.87** | **42.19** | 42.13 |
| | Multi-Doc. QA | 44.08 | 42.02 | **42.05** | 43.11 | 42.67 | 43.80 | **44.21** | 43.55 | **43.91** |
| | Summarization | 29.23 | 22.06 | **22.57** | 23.79 | **24.33** | 25.23 | **25.36** | 26.24 | **26.64** |
| | Few-shot | 69.24 | **62.78** | 62.32 | 67.20 | **67.28** | 68.10 | **68.62** | **68.84** | 68.67 |
| | Synthetic | 54.46 | 52.70 | **53.46** | 53.49 | **53.76** | 53.32 | **53.49** | 53.41 | **53.52** |
| | Code | 59.66 | 55.44 | **56.29** | 57.38 | **57.86** | 59.30 | **59.55** | 59.47 | **59.50** |
| | Ave. | 49.20 | 43.94 | **44.26** | 46.24 | **46.38** | 47.37 | **47.70** | 48.01 | **48.13** |
| Mistral-7B | Single-Doc. QA | 41.12 | 34.13 | **34.69** | **38.27** | 38.21 | **39.49** | 39.46 | 39.54 | **39.80** |
| | Multi-Doc. QA | 39.13 | 37.34 | **37.63** | 37.74 | **38.08** | **38.77** | 38.66 | **39.20** | 38.88 |
| | Summarization | 29.35 | 21.94 | **21.97** | 23.10 | **23.38** | 24.70 | **25.06** | 26.02 | **26.21** |
| | Few-shot | 70.57 | 63.33 | **66.46** | 68.10 | **68.27** | 69.51 | **70.07** | 70.13 | **70.49** |
| | Synthetic | 51.50 | 49.25 | **49.50** | 52.00 | **52.00** | 52.50 | **52.50** | 52.75 | **53.25** |
| | Code | 59.98 | 55.86 | **56.11** | 58.62 | **58.70** | 60.07 | **60.15** | **60.03** | 59.92 |
| | Ave. | 47.72 | 42.53 | **43.34** | 45.18 | **45.33** | 46.41 | **46.57** | 46.89 | **47.03** |

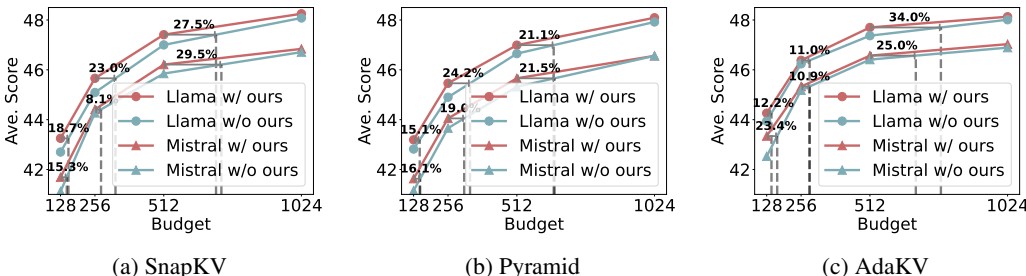

|     (a) SnapKV     |     (b) Pyramid     |     (c) AdaKV     |

Figure 1: Overview of Integrations. This demonstrates our algorithm achieves significant budget savings with the same score, providing 21.1-34% additional savings with base budget 512.

### 4.3 RESULTS

Tables 1, 2, and 3 present the quality scores across various task domains when integrating our algorithm into SnapKV, Pyramid, and AdaKV under different models and cache budgets $b$. Overall, our algorithm consistently improves post-eviction generation quality across all eviction methods. For instance, in Table 1, using LLaMA-3.1-8B, our algorithm significantly enhances SnapKV's quality scores across all 6 task domains at a budget size of 256, increasing the average score from 45.09 to 45.66. Similar improvements are observed with larger budgets, such as an increase from 47.00 to 47.41 at a budget of 512, and from 48.08 to 48.25 at a budget of 1024. Pyramid and AdaKV, built upon SnapKV, further optimize generation quality through inter-head budget allocation. However, due to predefined parameters, Pyramid shows diminished generalization performance on the two new LLMs, only slightly improving upon SnapKV at a budget of 128. In contrast, AdaKV, with its adaptive dynamic allocation strategy, maintains a quality advantage over SnapKV across both models. Although the optimizations from these two budget allocation algorithms vary, both show significant improvements across all budget sizes when combined with our algorithm.

Figure 1 shows the average scores across all 6 task domains. As the budget increases, generation quality improves for all methods, albeit with diminishing marginal returns. Interestingly, under these diminishing returns, our perturbation-constrained algorithm brings increasingly higher gains. For instance, in Figure 1a, when our algorithm is integrated with SnapKV at a budget of 512, the average score rises to 47.41. By interpolation, SnapKV alone would require a significantly larger cache budget (706) to achieve a similar quality score, implying a 27.5% savings with our algorithm. This is higher than the 23.0% savings at a budget of 256 and 18.7% at 128. Similar observations are also observed in Pyramid (Figure 1b) and AdaKV (Figure 1c). Thus, in real-world deployment, these significant savings in cache budgets yield substantial GPU memory savings, as well as improved decoding efficiency.

## 5 EMPIRICAL ANALYSIS OF PRACTICAL OUTPUT PERTURBATION

In this section, we empirically analyze the quality improvements of our method through practical output perturbation. Using the Qasper dataset, which consists of 200 single-document QA samples averaging 3,619 tokens, we compute the hidden states of the first decoding token on two base LLMs. For each sample, we record the hidden states of the first decoding token during inference and visualize the output perturbation compare to full KV cache case under different cache eviction methods.

### 5.1 HEAD-WISE REDUCTION IN OUTPUT PERTURBATIONS

Figure 2 visualizes the comparison of output perturbation between cache eviction methods with or without our algorithm across different attention heads. The results show that, in the majority of attention heads, our algorithm consistently yields lower output perturbation, whether applied to SnapKV or AdaKV, under both Llama-3.1-8B and Mistral-7B. For instance when integrating with SnapKV in Figure 2a and Figure 2b, our algorithm achieves lower output perturbation in 819 heads for Llama-3.1-8B and 748 heads for Mistral-7B out of the 1,024 attention heads. Thus, our algorithm could reduce output perturbation across the majority of attention heads, with an average reduction of 74.3% in four cases.

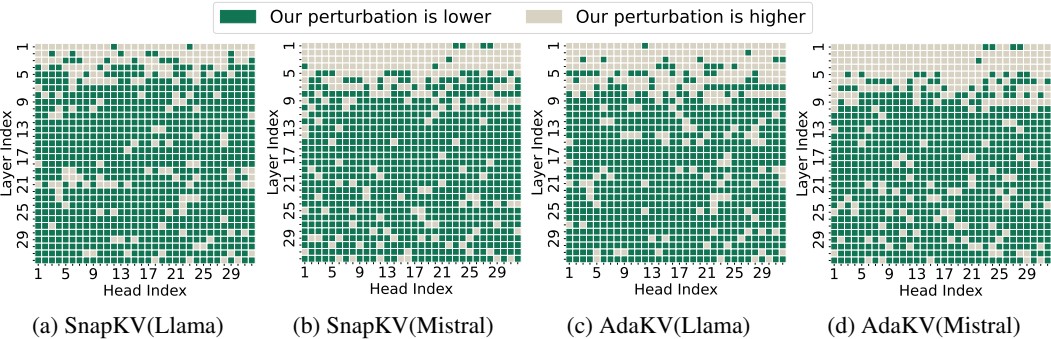

(a) SnapKV(Llama)  (b) SnapKV(Mistral)  (c) AdaKV(Llama)  (d) AdaKV(Mistral)

Figure 2: Comparison of head-wise perturbation.This compares output perturbation from cache eviction across attention heads between our algorithm and the previous attention weights based strategy. In four test cases, our algorithm reduces output perturbation in 74.3% of attention heads.

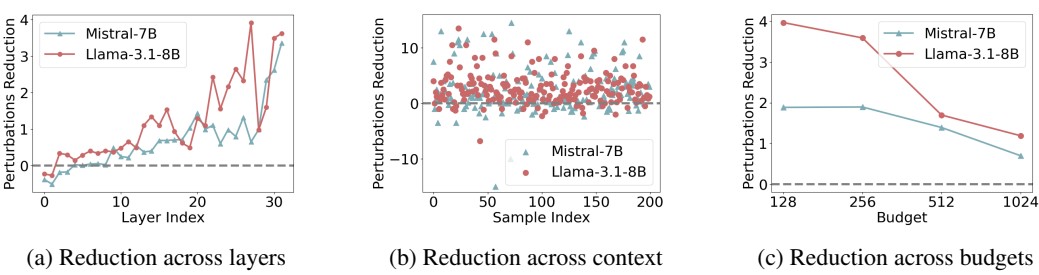

(a) Reduction across layers  (b) Reduction across context  (c) Reduction across budgets

Figure 3: Final-layer reduction in output perturbation

## 5.2 FINAL-LAYER REDUCTION IN OUTPUT PERTURBATIONS

Figure 3 presents the final-layer reduction in output perturbation when incorporating into SnapKV. A more comprehensive visual analysis, including application in AdaKV, can be found in Appendix B. As shown in Figure 3a, our algorithm gradually decreases perturbation layer by layer, with substantial reductions in the final layer. This gradual improvement is mainly due to the cumulative advantages across layers, leading to significantly lower output perturbation. Additionally, Figure 3b presents a detailed comparison of the final-layer output perturbation for different contexts in varying samples, which is directly related to the post-eviction generation token. These smaller perturbations in final layer output, directly leading to the generation being more aligned with the original context (using the full KV cache), explain the enhanced post-eviction generation quality. Figure 3c further summarizes the reduction in final perturbation across different budget sizes, demonstrating that our algorithm consistently produces smaller output perturbation regardless of budget size. In conclusion, our algorithm consistently reduces final-layer perturbation, underscoring its reliable superiority in identifying critical entries. This consistent advantage also explains its ability to enhance post-eviction output quality as observed in earlier experiments.

## 6 CONCLUSION

In this paper, we pinpoint a key limitation in current cache eviction methods: the reliance on intuitive heuristics of using attention weights to select critical cache entries for eviction. For the first time, we formalize the problem of critical cache entry selection from the perspective of output perturbation and provide a theoretical analysis. Furthermore, we propose a novel algorithm based on constraining output perturbation in the worst-case for critical cache selection, which is then integrated into existing SOTA cache eviction methods. Comprehensive evaluations using 16 datasets from Longbench demonstrate that our algorithm improves the performance of various cache eviction methods, across different task domains and budget constraints. Further empirical analysis also confirms and explains this benefit from the perspective of practical output perturbation: our algorithm consistently yields lower perturbation compared to previous methods that rely solely on attention weights, in all testings.

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

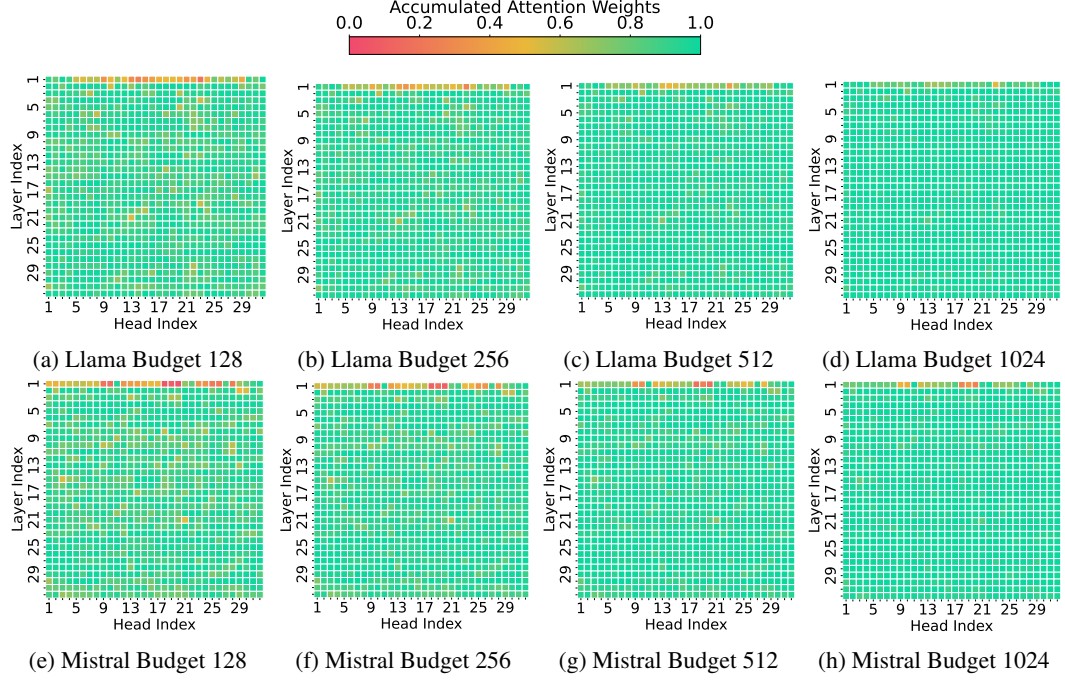

Figure 4: Accumulated attention weights.

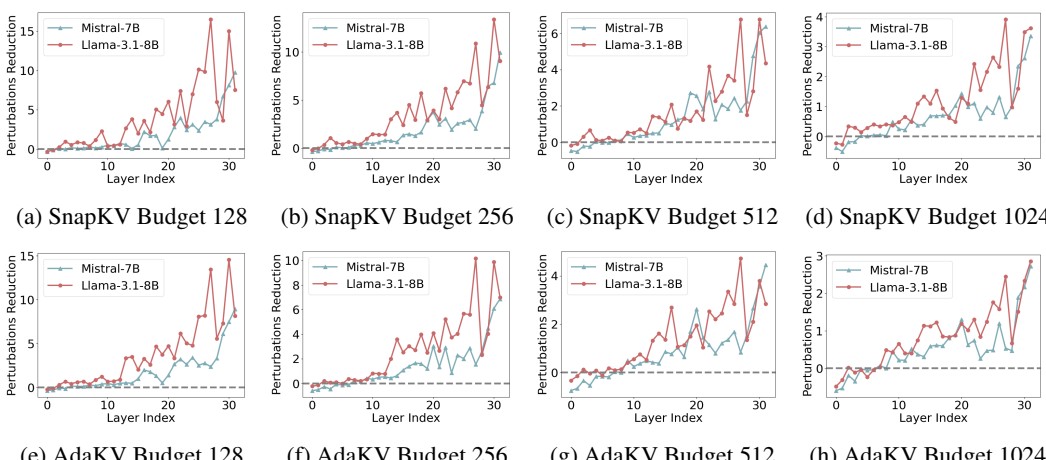

Figure 5: Perturbation reduction across layers.

## A  RELIABILITY OF ASSUMPTION 1

We ensure the reliability of Assumption 1 by analyzing the cumulative attention weights of critical KV Cache entries $\sum_{i=1}^{n} \mathcal{N}i Ai$ in individual heads with varying budgets. As shown in Figure 4, for varying models and budget levels, the majority of attention heads can effectively accumulate over 0.5 of the attention weights. The only exceptions are a few attention heads in the first layer. This is primarily due to the low sparsity of attention weights in certain heads of the first layer, a phenomenon that has been noted in many related studies Tang et al. (2024a); Zhang et al. (2024b;a). This observation aligns with the fact that certain heads in the first layer violate Assumption 1, which may lead our algorithm to increase the output perturbation of these heads as shown in Figure 2. However, this is not a significant issue, as the benefits from other attention heads in the subsequent layers accumulate, quickly offsetting these disadvantages.

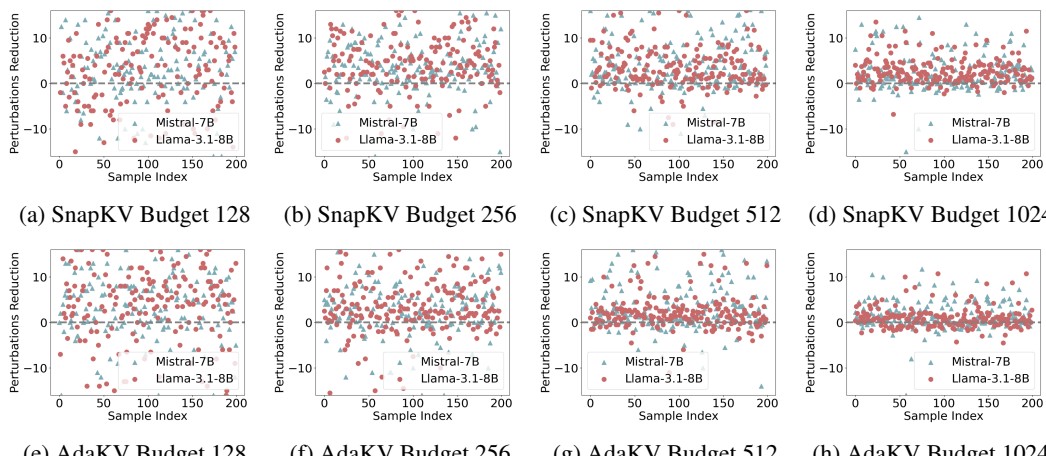

Figure 6: Perturbation reduction across contexts.

# B ADDITIONAL EMPIRICAL ANALYSIS

We provide a more detailed visual analysis to support the analysis conclusions drawn in the main paper. Figure 5 illustrates the process of reduced perturbation at each layer when combining AdaKV and SnapKV with our algorithm under different budgets. Notably, our algorithm demonstrates a more significant reduction in perturbation under low budgets, primarily due to the larger compression loss in this scenario, which creates substantial optimization space. A similar trend is also evident in the visualization of the perturbation reduction at the final layer across different contexts, as shown in Figure 6. Overall, it is clear that in all cases, whether consider acrossing different layers or contexts, our algorithm significantly reduces the output perturbation after cache eviction. This ultimately leads to lower quality loss following cache eviction and correspondingly enhances the final generation quality.

# C DETAILS OF 16 DATASETS

Table 4 provides detailed information on the 16 datasets in LongBench.

Table 4: Details of 16 Datasets

| Task | Task Type | Eval metric | Avg len | Language | Sample Num |
|---|---|---|---|---|---|
| NarrativeQA | Single-Doc. QA | F1 | 18,409 | EN | 200 |
| Qasper | Single-Doc. QA | F1 | 3,619 | EN | 200 |
| MultiFieldQA-en | Single-Doc. QA | F1 | 4,559 | EN | 150 |
| HotpotQA | Multi-Doc. QA | F1 | 9,151 | EN | 200 |
| 2WikiMultihopQA | Multi-Doc. QA | F1 | 4,887 | EN | 200 |
| MuSiQue | Multi-Doc. QA | F1 | 11,214 | EN | 200 |
| GovReport | Summarization | Rouge-L | 8,734 | EN | 200 |
| QMSum | Summarization | Rouge-L | 10,614 | EN | 200 |
| MultiNews | Summarization | Rouge-L | 2,113 | EN | 200 |
| TREC | Few-shot Learning | Accuracy | 5,177 | EN | 200 |
| TriviaQA | Few-shot Learning | F1 | 8,209 | EN | 200 |
| SAMSum | Few-shot Learning | Rouge-L | 6,258 | EN | 200 |
| PassageCount | Synthetic | Accuracy | 11,141 | EN | 200 |
| PassageRetrieval-en | Synthetic | Accuracy | 9,289 | EN | 200 |
| LCC | Code | Edit Sim | 1,235 | Python/C#/Java | 500 |
| RepoBench-P | Code | Edit Sim | 4,206 | Python/Java | 500 |

Table 5: Ablation Study of $\alpha$ in SnapKV Integration

| SnapKV | | 6711 Full Cache | $b = 128$ ablation | $b = 128$ w/ ours | $b = 256$ ablation | $b = 256$ w/ ours | $b = 512$ ablation | $b = 512$ w/ ours | $b = 1024$ ablation | $b = 1024$ w/ ours |
|---|---|---|---|---|---|---|---|---|---|---|
| Llama-3.1-8B | Single-Doc. QA | 43.80 | 35.17 | **35.56** | 37.99 | **38.34** | 41.02 | **41.49** | 42.59 | **42.75** |
| | Multi-Doc. QA | 44.08 | 40.78 | **41.80** | 43.36 | 42.79 | 43.82 | 43.51 | 44.20 | 43.87 |
| | Summarization | 29.23 | **22.34** | 22.18 | 23.78 | **24.02** | 25.32 | **25.35** | 26.63 | 26.47 |
| | Few-shot | 69.24 | **60.24** | 60.00 | 63.95 | 63.91 | 67.22 | 66.99 | 68.51 | 68.51 |
| | Synthetic | 54.46 | **53.23** | 52.48 | 53.53 | **54.51** | 53.76 | **54.49** | 53.15 | 54.14 |
| | Code | 59.66 | 53.84 | **54.22** | 57.17 | **57.20** | 58.69 | **58.75** | 59.63 | 59.45 |
| | Ave. | 49.20 | 43.11 | **43.25** | 45.54 | **45.66** | 47.31 | **47.41** | 48.21 | **48.25** |
| Mistral-7B | Single-Doc. QA | 41.12 | 33.68 | **33.92** | 37.96 | **38.17** | 40.11 | **40.42** | 39.93 | 39.63 |
| | Multi-Doc. QA | 39.13 | 36.15 | **36.66** | 37.14 | **37.34** | 38.25 | **38.36** | 39.23 | **39.40** |
| | Summarization | 29.35 | 21.73 | **21.85** | 23.21 | **23.24** | 24.77 | 24.71 | 26.12 | **26.18** |
| | Few-shot | 70.57 | 60.45 | **60.53** | 66.04 | 66.02 | 68.79 | 68.44 | 70.03 | 69.97 |
| | Synthetic | 51.50 | **49.75** | **49.75** | 50.50 | **50.75** | 51.75 | **52.25** | 52.00 | 52.00 |
| | Code | 59.98 | 54.34 | **54.35** | 57.34 | **57.39** | 59.16 | **59.54** | 59.84 | **59.93** |
| | Ave. | 47.72 | 41.51 | **41.69** | 44.30 | **44.41** | 46.10 | **46.21** | **46.85** | 46.84 |

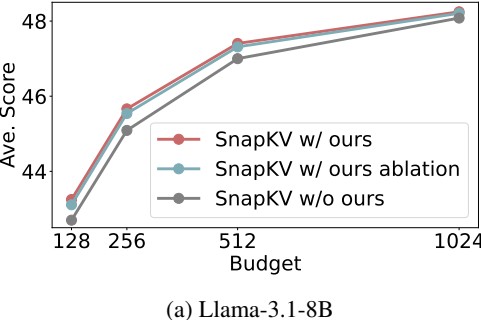

(a) Llama-3.1-8B

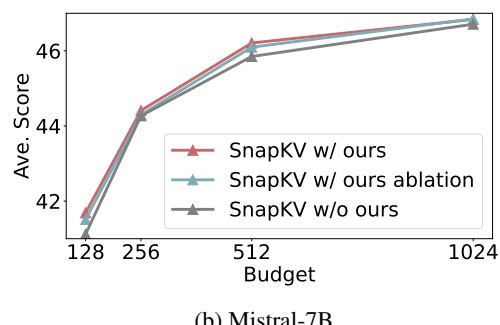

(b) Mistral-7B

Figure 7: Ablation Study of $\alpha$ in SnapKV Integration

## D ABLATION STUDY OF STAGE 1 IN ALGORITHM 1

In our algorithm, we initially set $\alpha = 0.25$ to allocate a portion of the budget for collecting high attention weights, corresponding to Assumption 1, to ensure the validity of Theorem 3. We further conducted ablation experiments by setting $\alpha = 0$, removing this mechanism, to demonstrate the necessity of budget splitting in guaranteeing the validity of Theorem 3. Table 5 shows the results of ablating $\alpha$. As seen, when $\alpha$ is set to 0, both Mistral-7B and Llama-3.1-8B exhibit a general quality decline. This is primarily because, for some samples, the cumulative attention weights may fall below 0.5, causing the algorithm to lose the alignment with minimizing input perturbation in the worst-case scenario, which in turn negatively impacts performance. However, as shown in Figure 7, despite the performance drop in the ablation version, it still outperforms the original SnapKV. This is because, even with $\alpha$ set to 0, the selection algorithm in most samples of datasets can still aggregate enough attention weights to constrain the output perturbation, resulting in performance gains. This also highlights, to some extent, the robustness of our algorithm.

## E NEEDLE-IN-A-HAYSTACK TEST

Our approach is further evaluated using the widely adopted synthetic data benchmark, Needle-in-a-Haystack, to assess its enhancement of existing cache eviction methods in long-text retrieval tasks. This benchmark involves embedding a crucial sentence (the "needle") within a lengthy context (the "haystack"), subsequently measuring the model's ability to retrieve this specific sentence from the document. The x-axis represents the document's context length, while the y-axis denotes the needle's insertion depth. The Average Score is calculated by averaging retrieval performance across

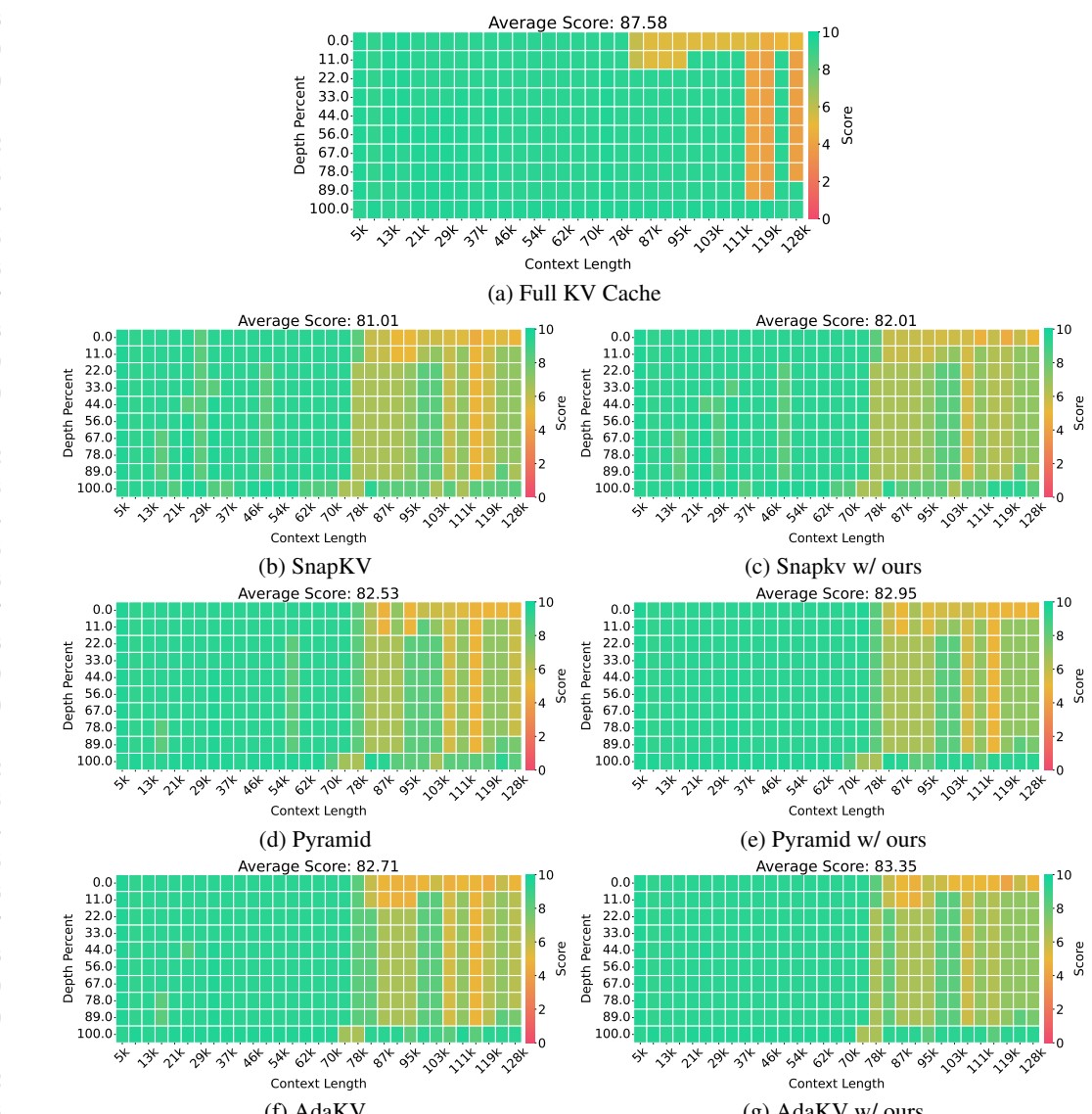

Figure 8: Needle-in-a-Haystack Test on Llama-3.1-8B with budget 128.

different context lengths, with higher scores signifying an improved ability of the model for effective contextual retrieval.

As shown in Figure 8, we employ three cache eviction algorithms under a budget of 128, testing with Llama-3.1-8B to reach its maximum supported length of 128K, and calculate their respective scores. Results demonstrate a progressive increase in retrieval scores among the original cache eviction methods, from SnapKV, Pyramid, to AdaKV showing gradual improvements in retrieval performance. Notably, when these methods were combined with our algorithm, all experienced accuracy enhancements. Interestingly, the combined scores across the three methods showed a similar pattern of progressive retrieval improvement from SnapKV to Pyramid and then to AdaKV. This indicates that our algorithm is compatible with a range of cache eviction methods and, when paired with stronger cache eviction techniques, it can further amplify retrieval capabilities.

# F ANALYSIS OF PREVIOUS SOLELY ATTENTION WEIGHTS-BASED SELECTION FROM A PERTURBATION PERSPECTIVE

Our algorithm differs from the previous solely attention weights-based selection method primarily in Stage 2. Specifically, by modifying stage 2 of our algorithm to perform the same attention weights-based selection operation as in stage 1, our approach will degrade into the previous method. This modification allows us to conveniently apply perturbation-constrained theory to analyze the earlier attention weights-based selection strategy.

**Theorem 4.** *Previous solely attention weights-based selection is equivalent to minimizing another upper bound $\hat{\theta}^{relax}$, a relaxed form of $\hat{\theta}$, with remaining budget $b''$ based on stage 1 selection.*

$$\hat{\theta}^{relax} = C' - M\left(2 - \frac{1}{\sigma}\right)\sum_{i=1}^{n} \mathcal{N}_i'' A_i \quad where \quad M = MIN(\|\boldsymbol{\mathcal{V}}_{i,:}\|_1) \tag{18}$$

*Proof.* We relax the upper bound $\hat{\theta}$ by utilizing $M = MIN(\|\boldsymbol{\mathcal{V}}_{i,:}\|_1)$:

$$\hat{\theta} = C' - \left(2 - \frac{1}{\sigma}\right)\sum_{i=1}^{n} \mathcal{N}_i'' A_i \|\boldsymbol{\mathcal{V}}_{i,:}\|_1 \leq C' - M\left(2 - \frac{1}{\sigma}\right)\sum_{i=1}^{n} \mathcal{N}_i'' A_i = \hat{\theta}^{relax} \tag{19}$$

In the solely attention weights-based selection strategy, the $\mathcal{N}''$ selection is performed using $Top - K(A_i, b'')$ to maximize $\sum_{i=1}^{n} \mathcal{N}_i'' A_i$. This is therefore equivalent to minimizing the relaxed upper bound, $\hat{\theta}^{relax}$.

$\square$

Theorem 4 demonstrates that the solely attention weights-based selection strategy is equivalent to minimizing the relaxed upper bound $\hat{\theta}^{relax}$. In contrast, our algorithm optimizes a tighter upper bound, $\hat{\theta}$. While this does not guarantee that our approach will yield a strictly better solution, intuitively, an algorithm designed to optimize a tighter bound often achieves better results. Theorem 4 also provides some insight into why a critical KV Cache subset can replace the entire KV Cache in cache eviction methods. Due to the power-law distribution of attention weights Zhang et al. (2024b), removing most cache entries with near-zero attention weights has a negligible impact on this upper bound. Consequently, the perturbation to the actual output is also bounded by this upper bound.

# G CHOICE OF DISTANCE METRIC

In this paper, we use the $L_1$ distance as the simplest distance metric for our analysis, while future work could explore the use of $L_2$ distance or other metrics. We chose $L_1$ distance for the following two reasons: 1. Theoretical Perspective: The $L_1$ distance, as a straightforward metric, facilitates the construction and derivation of our theoretical framework. 2.Practical Perspective: Considering that BF16 (half-precision floating-point) is commonly used in inference computations, the $L_1$ distance operations derived from our algorithm provide better numerical precision. In contrast, metrics such as $L_2$ distance may introduce numerical precision issues due to the squaring and square-rooting operations inherent in their computation. Fundamentally, the choice of distance metric is orthogonal to our primary contributions. Future works can further investigate the impact of different distance metrics.

# H COMPARISION TO STREAMING LLM

We include the results of StreamingLLM, an early non-selective cache eviction method, for comparative analysis. This method retains only several initial cache entries along with the most recent ones within a sliding window. Detailed results for StreamingLLM (SLM) on the LongBench benchmark are presented in Table 6, evaluated across various LLM configurations and cache budget settings. The non-selective cache eviction strategy employed by StreamingLLM leads to considerable information loss, resulting in significantly lower post-eviction quality compared to selection-based cache

Table 6: Comparison between StreamingLLM and Selection-Based Cache Eviction (e.g., SnapKV w/ Ours)

| SnapKV | | 6711 Full Cache | b = 128 | | b = 256 | | b = 512 | | b = 1024 | |
|---|---|---|---|---|---|---|---|---|---|---|
| | | | SLM | Ours | SLM | Ours | SLM | Ours | SLM | Ours |
| Llama-3.1-8B | Single-Doc. QA | 43.80 | 24.94 | **35.56** | 26.10 | **38.34** | 28.67 | **41.49** | 30.48 | **42.75** |
| | Multi-Doc. QA | 44.08 | 35.37 | **41.80** | 35.18 | **42.79** | 35.22 | **43.51** | 36.66 | **43.87** |
| | Summarization | 29.23 | 19.43 | **22.18** | 20.82 | **24.02** | 22.90 | **25.35** | 24.37 | **26.47** |
| | Few-shot | 69.24 | 54.58 | **60.00** | 57.98 | **63.91** | 62.53 | **66.99** | 64.94 | **68.51** |
| | Synthetic | 54.46 | **53.75** | 52.48 | 53.35 | **54.51** | 51.99 | **54.49** | 48.24 | **54.14** |
| | Code | 59.66 | 52.16 | **54.22** | 54.54 | **57.20** | 56.17 | **58.75** | 57.62 | **59.45** |
| | Ave. | 49.20 | 38.42 | **43.25** | 39.75 | **45.66** | 41.52 | **47.41** | 42.56 | **48.25** |
| Mistral-7B | Single-Doc. QA | 41.12 | 23.51 | **33.92** | 25.25 | **38.17** | 26.51 | **40.42** | 27.89 | **39.63** |
| | Multi-Doc. QA | 39.13 | 29.66 | **36.66** | 29.82 | **37.34** | 30.60 | **38.36** | 31.50 | **39.40** |
| | Summarization | 29.35 | 18.21 | **21.85** | 19.43 | **23.24** | 21.01 | **24.71** | 23.79 | **26.18** |
| | Few-shot | 70.57 | 57.12 | **60.53** | 61.14 | **66.02** | 65.34 | **68.44** | 67.56 | **69.97** |
| | Synthetic | 51.50 | 42.00 | **49.75** | 42.25 | **50.75** | 43.25 | **52.25** | 43.75 | **52.00** |
| | Code | 59.98 | 51.04 | **54.35** | 54.46 | **57.39** | 56.11 | **59.54** | 58.28 | **59.93** |
| | Ave. | 47.72 | 35.72 | **41.69** | 37.52 | **44.41** | 39.47 | **46.21** | 41.02 | **46.84** |

eviction methods. For instance, consider SnapKV w/ ours on Llama3.1-8B as a representative example of selection-based cache eviction approaches. The quality scores of StreamingLLM under cache budgets ranging from 128 to 1024 are 38.42, 39.75, 41.52, and 42.56, respectively. In contrast, the selection-based cache eviction methods achieve significantly higher scores of 43.25, 45.66, 47.41, and 48.25 under the same budget configurations.

## I  THE RELATIONSHIP BETWEEN THE ATTENTION OUTPUT PERTURBATION AND THE GENERATION QUALITY

In LLM computations, the attention output serves as an input to the FeedForward Neural Network (FFN) module, producing outputs that are subsequently passed to the Language Model (LM) head to generate the token vocabulary distribution. Our algorithm specifically aims to reduce perturbations in attention outputs caused by cache eviction methods. This reduction lowers the perturbations in the inputs to downstream network components (FFN and LM head), thereby mitigating perturbations in the final token vocabulary distribution. Consequently, our method reduces the impact of cache eviction on output token generation, which is critical to maintaining high-quality generation results.

This conclusion is also supported both theoretically and empirically. Theoretically, the relationship between reduced input perturbations and diminished output variations is well-established, as seen in Lipschitz continuity Xu & Zhang (2024); Khromov & Singh (2024), which posits that functions with bounded Lipschitz constants exhibit smaller output differences for smaller input differences. This principle is consistent with our approach to minimizing attention output perturbations, thereby ensuring subsequent generated tokens. Similarly, FFN pruning techniques in LLMs Dong et al. (2024); Sun et al. (2024b) also demonstrate the practical success of minimizing perturbations to downstream layers' outputs, further validating our strategy.

## J  ADDITIONAL RELATED WORKS AND FUTURE DIRECTIONS

Sparse attention methods Jiang et al. (2024); Tang et al. (2024b); Lv et al. (2024) are conceptually related to the KV cache eviction methods discussed in this paper. While KV cache eviction retains only a small subset of essential KV cache entries, sparse attention methods maintain all entries during inference. However, during computation, only the most critical entries are selectively utilized in the sparse attention mechanism. Consequently, sparse attention methods do not reduce the memory footprint of the KV cache but enhance inference speed and often offer better output quality than cache eviction methods Tang et al. (2024b). Existing sparse attention methods typically rely on approximate estimations of attention weights to identify critical entries Tang et al. (2024b); Lv et al.

Table 7: AlpacaEval Benchmark (Llama-3.1-8B)

| Method | Alpaca Eval 2.0 | |
|---|---|---|
| | Win Rate | LC Win Rate |
| Evaluator=DeepSeek-Chat | | |
| Full KV Cache | 19.84 | 16.71 |
| SnapKV w/o ours | 13.63 | 11.10 |
| SnapKV w/ ours | **14.46** | **12.05** |
| Evaluator=Yi-Large | | |
| Full KV Cache | 25.72 | 24.34 |
| SnapKV w/o ours | 19.61 | 18.76 |
| SnapKV w/ ours | **21.20** | **19.79** |

Table 8: LongBench Evaluation on Qwen2.5-32B-Instruct (SnapKV)

| | SnapKV | 6711 Full Cache | $b=128$ | | $b=256$ | | $b=512$ | | $b=1024$ | |
|---|---|---|---|---|---|---|---|---|---|---|
| | | | w/o ours | w/ ours | w/o ours | w/ ours | w/o ours | w/ ours | w/o ours | w/ ours |
| Qwen2.5-32B | Single-Doc. QA | 42.23 | 32.34 | **32.87** | **37.68** | 37.38 | 39.50 | **40.23** | 41.30 | **41.63** |
| | Multi-Doc. QA | 54.47 | 48.14 | **48.29** | 52.02 | **52.06** | 53.28 | **53.68** | 53.83 | **54.53** |
| | Summarization | 25.45 | 18.62 | **19.21** | 20.48 | **20.86** | 22.31 | **22.60** | 23.48 | **23.56** |
| | Few-shot | 66.48 | 51.02 | **52.25** | 59.11 | **59.95** | 64.21 | **64.58** | **66.10** | 65.92 |
| | Synthetic | 55.25 | 53.17 | **54.34** | **54.21** | 53.97 | 54.62 | **54.75** | 54.53 | **54.75** |
| | Code | 40.39 | 36.51 | **37.06** | 38.31 | **38.57** | 39.44 | 39.32 | **39.47** | 39.36 |
| | Ave. | 47.32 | 39.36 | **40.04** | 43.31 | **43.49** | 45.38 | **45.71** | 46.38 | **46.57** |

(2024). Future works could explore integrating our proposed perturbation-constrained selection algorithm to refine these methods by achieving more accurate critical cache entry identification.

Some adaptive methods in KV cache eviction or sparse attention, such as Ge et al. (2024b); Jiang et al. (2024), employ varying critical cache selection strategies tailored to the characteristics of different attention heads. For example, some heads use attention weights based selection, while others utilize fixed patterns, such as recent window-based or special token-based approaches. Our method can also be applied to enhance performance in the head which according to attention weights-based selection strategies, providing a boost to adaptive methods.

Perturbation-based analysis has achieved remarkable success in the field of neural network interpretability. For instance, Catformer Davis et al. (2021) leverages output perturbation analysis to design more stable network architectures, while Admin Liu et al. (2020) examines the amplification of output perturbations in residual blocks to propose improved training schemes. In this paper, we analyze the output perturbations caused by cache eviction within the attention mechanism, leading to the design of more effective critical cache selection metrics. From the perspective of perturbation analysis, different works focus on the perturbation in various locations, such as residual connections and attention mechanisms. Future research could combine these perturbation analysis strategies to examine network perturbations in greater detail, guiding the design of more robust network architectures, training methodologies, and inference schemes.

## K    EVALUATION ON ALPACAEVAL BENCHMARK

To evaluate our algorithm on a wider spectrum of scenario, we further test the performance of SnapKV and SnapKV with ours using Llama-3.1-8B in AlpacaEval Li et al. (2023); Dubois et al. (2024), an automatic evaluation framework designed to assess the performance of instruction-following. For automatic annotations, we employed Deepseek-chat Bi et al. (2024) and Yi-Large [1] as Auto-annotators. Regarding compression settings, we analyze AlpacaEval and found that the average input length is 36.5 tokens, while the average generated length is 567 tokens. Thus, we conduct compression experiments by compressing the cache size to 64 tokens when the context length

---

[1]https://platform.lingyiwanwu.com/

exceeded 256 tokens—approximately half of the average generated length. As shown in Table 7, our enhanced version of SnapKV significantly outperforms the original version. For instance, using Yi-Large as the Auto-annotator, our method improves the scores for Win Rate and LC Win Rate from 19.61 and 18.76 to 21.20 and 19.79, respectively. This demonstrates the versatility and applicability of our algorithm across a broader range of scenarios.

## L  RESULTS ON LARGER-SCALE LLMS

We conduct further experiments on larger-scale LLMs, specifically Qwen2.5-32B-Instruct Team (2024), as shown in Table 8. SnapKV was evaluated both with and without our proposed algorithm under various budget constraints, with our method consistently delivering improved quality scores across different budget settings.

