# OpenReview forum: "Identify Critical KV Cache in LLM Inference from an Output Perturbation Perspective"
_ICLR.cc/2025/Conference — Submitted to ICLR 2025_

### Official Review · Reviewer_nKeS · 2024-10-20

**Soundness:** 2
**Presentation:** 2
**Contribution:** 3
**Rating:** 6
**Confidence:** 4

**Summary:**

The paper proposes a novel method to identify critical KVs in the KVCache for effective KV eviction. Unlike previous methods that solely rely on the attention score, the paper focuses on the attention output perturbation instead. It comes up with a theoretical upper bound for the attention output $L_1$ perturbation to optimize upon. More specifically, the paper proposes to include a small faction of KVs in the critical KVCache to capture the attention score while the rest to capture the derived upper bound term $\sum_{i=1}^n N_i A_i \Vert V_{i, :}\Vert_1$. Empirically, the paper shows that combined with this critical KV identification algorithm, they can consistently outperform the base version of SnapKV, Pyramid, and AdaKV on LongBench and over LLaMA-3.1-8B and Mistral-7B.

**Strengths:**

1. The KVCache compression problem is indeed well-motivated under the long-context scenario. Conventional approaches usually identify critical KVs based on the heuristic that tokens with a higher attention score are more important. In contrast, this work proposes a novel term derived from the attention output perturbation to capture the importance of different KVs with theoretical guarantees.

2. The proposed method can be easily integrated into scenarios requiring critical KV identification, such as KV Compression or sparse-attention serving systems.

3. The additional ablation studies further complement its empirical evaluations and the conclusions that the proposed method can reduce attention output perturbations compared to the heuristics-based approaches.

**Weaknesses:**

1. The notations and writing for the mathematical derivations are a bit confusing (see the question section for details.), which is the key contribution of this work and thus needs to be improved to give a clearer and more rigorous proof.

2. Empirically, the experiments focus mostly on small models around 7B. Some results on a larger model (e.g., ~70B) would be more convincing. Additionally, the paper can add some needle-in-the-haystack-type tasks to further show the effectiveness of the proposed method in terms of critical KV identification. One of my concerns for the general eviction-based sparse attention approach is that in a multi-turn or complex reasoning task, the evicted KVs can be important to a later decoding step.

**Questions:**

1. In Eq. (6), $L$ is defined to be the $L_1$ norm of the attention output deviation. However, in Eq. (12) - Eq. (15), $L$ directly equals the derived upper bound for the attention output deviation. This is a bit confusing and seems to be a typo to me as the author should either use $L \leq$ or $\theta =$ for the derived upper bounds instead of $L=$ for Eq. (12) - Eq. (15).

2. If I understand Theorem 3 correctly, the author is actually trying to claim $\arg\max_{N}\gamma = \arg\min_{N}\theta$ instead of $\max_{N}\gamma = \min_{N}\theta$, a simple counter-example is when $\sum_i^{n}N_i A_i = 1$ (the selected tokens can capture all the mass), in which case, $\min_{N}\theta=0$ and $\max_{N}\gamma>0$.

3. The proof of Theorem 3 is not quite straightforward for me. Firstly, the proposed inequality $\sum_i^n N_i A_i > \sum \text{Top}_k (A, b') > 0.5$ is not guaranteed for an arbitrary set of $N_i$ even if we have assumption 1. As we can selectively choose $N_i$ to not include tokens from $\text{Top}_k (A, b')$, in which case this is not guaranteed. Secondly, even if $\sum_i^n N_i A_i > 0.5$ is guaranteed, the derivation of $\arg\max_N \gamma = \arg\min_N \theta$ is not quite straightforward. More specifically, the assumption that
$(2-1/\sum_i^n N_i A_i)>0$  is not a sufficient condition to derive that $\arg\max_N\gamma = \arg\min_N\theta$. For instance, I can have two sets of $N_i$, the first set has $\sum_i N_i A_i = 0.51$ and the second set has $\sum_i N_i A_i = 0.99$, then I can adversarially set $\Vert V_i\Vert_1$ so that the $\gamma$ corresponds to the first set of $N_i$ is slightly higher than that of the second set of $N_i$. According to your claim, you would choose the first set of $N_i$, which gives you a slightly higher $\gamma$ but way smaller $2-1/\sum_i^n N_i A_i$ in which case $\arg\max_N \gamma \neq \arg\min_N \theta$.  Please correct me if I am wrong on this.

Suggestions on 3: You might want to move your algorithm description before stating the theorem as I am guessing the proposed inequality $\sum_i^n N_i A_i > \sum \text{Top}_k (A, b') > 0.5$ is based on your algorithm rather than for arbitrary $N_i$. And for theorem 3, if you want to prove the same bound, I would suggest you modify your assumption 1 to be $>0.51$ rather than $>0.5$ then you can lower bound the $2-1/\sum_i^n N_i A_i$ term with $2-\frac{1}{0.51}$ and can further upper bound the current $\theta$ with $\hat{\theta}=C-(2-\frac{1}{0.51}) \gamma$ and reach to the conclusion that $\arg\max_N\gamma = \arg\min_N\hat{\theta}$.

4. Empirical-wise, if time allows I would be happy to see some results for a larger LLM (e.g., 70B) as well as for some needle-in-the-haystack type of task.

In general, the problem is well-motivated, and the proposed idea with theoretical support is novel. I would be happy to raise the score if the authors can address my concerns during the rebuttal period.

---

> ### Author Response · Authors · 2024-11-22
> **Reply to Reviewer nKeS**
>
> We apologize for any confusion caused. Your suggestions are highly valuable, and we have revised our manuscript to address unclear statements. These changes have greatly enhanced the clarity of our work. Thank you for your meaningful contribution!
>
> ## **Reply to Question1 and Question2：**
>
> Thank you for pointing out the typos, and we apologize for any confusion caused. We have corrected these issues in the revised version.
>
>
> ## **Reply to Question 3：**
>
> Yes, you are right. Thanks for pointing out the imprecise wording.
> We have revised the paper by moving the algorithm section to precede the theoretical analysis, improving the overall readability. Additionally, we have provided detailed proof for Theorem 3, which has significantly enhanced the presentation of our work.
>
>
> ## **Reply to Question4 Needle-in-the-Haystack Results：**
>
> Thanks for your suggestions! We have added the needle-in-the-haystack test, with detailed results included in Appendix E of our paper. Using Llama-3.1-8B with a budget of 128, we extended inference up to the model’s maximum supported length of 128K. Under the Full Cache setting, the retrieval score reached 87.58. Our algorithm enhances the long-context retrieval capabilities of all cache eviction methods, including SnapKV, Pyramid, and AdaKV, improving their retrieval scores from 81.01, 82.53, and 82.71 to 82.01, 82.95, and 83.35, respectively.
>
> | Needle-in-the-Haystack (128K) 128 Budget | Full Cache | w/o Ours | w/ Ours   |
> |-------------|------------|----------|-----------|
> | SnapKV      | 87.58      | 81.01    | **82.01** |
> | Pyramid     | 87.58      | 82.53    | **82.95** |
> | AdaKV       | 87.58      | 82.71    | **83.35** |
>
>
> ## **Reply to Question4 Larger LLM Results：**
>
>
> Thanks for your suggestions! However, we found that for 70B LLMs, even with cache compression, running long-sequence benchmarks demands immense computational resources. This is primarily due to the significant computational overhead during the prefill stage for long sequences and the substantial GPU memory consumption resulting from the model's large parameter size. We believe this may also explain why most cache compression studies typically evaluate on small parameter sizes like 7B or 8B.
>
> To address your concerns within our feasible resource constraints, we opted to evaluate another model, Qwen-2.5-32B, a leading open-source LLM. We assessed SnapKV, both with and without our algorithm, under budgets of 128 and 1024.
> With the discussion phase already halfway through, we look forward to engaging with all reviewers promptly to resolve concerns. Thus, we present experimental results have achieved so far, with the complete experimental findings to be included in the final version of the paper.
>
>
>
> | 128 Budget          | Single-Doc. QA | Multi-Doc. QA | Summarization | Few-shot  | Synthetic | Code      | Ave.      |
> |----------------------|----------------|---------------|---------------|-----------|-----------|-----------|-----------|
> | Qwen2.5-32B w/o ours | 32.34          | 48.14         | 18.62         | 51.02     | 53.17     | 36.51     | 39.36     |
> | Qwen2.5-32B w/  ours | **32.87**      | **48.29**     | **19.21**     | **52.25** | **54.34** | **37.06** | **40.04** |
>
>
>
> | 512 Budget          | Single-Doc. QA | Multi-Doc. QA | Summarization | Few-shot  | Synthetic | Code | Ave. |
> |----------------------|----------------|---------------|---------------|-----------|-----------|------|------|
> | Qwen2.5-32B w/o ours | 39.50          | 53.28         | 22.31         | 64.21     | -         | -    | -    |
> | Qwen2.5-32B w/  ours | **40.23**      | **53.68**     | **22.60**     | **64.58** | -         | -    | -    |
>
>
> | 1024 Budget         | Single-Doc. QA | Multi-Doc. QA | Summarization | Few-shot  | Synthetic | Code      | Ave.      |
> |----------------------|----------------|---------------|---------------|-----------|-----------|-----------|-----------|
> | Qwen2.5-32B w/o ours | 41.30          | 53.83         | 23.48         | **66.10** | 54.53     | **39.47** | 46.38     |
> | Qwen2.5-32B w/  ours | **41.63**      | **54.53**     | **23.56**     | 65.92     | **54.75** | 39.36     | **46.57** |
>
> We hope our responses have helped address your concerns. Should you have any additional questions, we would be more than happy to provide further clarification.

---

> > ### Comment · Reviewer_nKeS · 2024-11-22
> >
> > The updated version of the paper addresses most of my concerns, with a clearer presentation and additional empirical evidence to support its methodology. I have raised my score accordingly.

---

> > > ### Author Response · Authors · 2024-11-25
> > > **Thanks for Your Timely Reply**
> > >
> > > We sincerely appreciate your timely response and recognition of our work.  We have incorporated additional results for Qwen2.5-32B-Instruct in Appendix L of our manuscript, demonstrating that our method achieves quality improvements across all budget settings. Once again, thank you for your valuable efforts in reviewing our work, which have greatly contributed to its improvement!
> > >
> > >
> > > | 256 Budget          | Single-Doc. QA | Multi-Doc. QA | Summarization | Few-shot  | Synthetic | Code      | Ave.      |
> > > |----------------------|----------------|---------------|---------------|-----------|-----------|-----------|-----------|
> > > | Qwen2.5-32B w/o ours | **37.68**      | 52.02         | 20.48         | 59.11     | **54.21** | 38.31     | 43.31     |
> > > | Qwen2.5-32B w/  ours | 37.38          | **52.06**     | **20.86**     | **59.95** | 53.97     | **38.57** | **43.49** |
> > >
> > >
> > > | 512 Budget         | Single-Doc. QA | Multi-Doc. QA | Summarization | Few-shot  | Synthetic | Code      | Ave.      |
> > > |----------------------|----------------|---------------|---------------|-----------|-----------|-----------|-----------|
> > > | Qwen2.5-32B w/o ours | 39.50          | 53.28         | 22.31         | 64.21     | 54.62     | **39.44** | 45.38     |
> > > | Qwen2.5-32B w/  ours | **40.23**      | **53.68**     | **22.60**     | **64.58** | **54.75** | 39.32     | **45.71** |

---

### Official Review · Reviewer_19en · 2024-10-29

**Soundness:** 3
**Presentation:** 2
**Contribution:** 3
**Rating:** 6
**Confidence:** 4

**Summary:**

This paper proposes a novel method to identify essential KV cache values based on a perturbation-constrained selection algorithm. Empirical results highlight this approach's performance in small KV budgets against other KV-cache eviction methods.

**Strengths:**

This paper introduces a novel perturbation-constrained selection algorithm. The criteria underlying the approach are well-motivated and rooted in mathematical principles with proof and derivation. The algorithm’s design is intuitive and straightforward, facilitating easy integration into existing frameworks.

**Weaknesses:**

While the overall paper is relatively sound, there are some writing and formatting issues I would like to see addressed:

* Section 3.2: In my opinion, this section needs to be reformatted and partially rewritten to be more concise with clearer structure. As the most essential part of the paper, it is currently challenging to read, with many notations introduced ambiguously and references placed somewhat haphazardly.
* Equation Alias: The author should create an alias for $\frac{N \cdot A}{\sum_n N_i A_i}$ to simplify and streamline the equations that reference this expression.
* Assumption 1 and Theorem 3: I believe these should form their own subsection, as they represent the core approach, while the other sections primarily provide proofs and background information.

Additionally, there are several points I still find unclear, and I would appreciate clarification from the authors:

* Assumption 1 Interpretation: I am having difficulty reconciling the assumptions made in Assumption 1 with several statements in the paper:

> Lines 17–19: "Yet they rely... attention weights"

> Lines 60–63: "Although the term... higher attention weights"

> Lines 128–129: "However... critical entries"

These statements critique the empirical reliance on attention weights as indicators of importance. Could the authors clarify how they reconcile these criticisms with their own assumption about using attention scores to represent the importance of key-value entries?

* Arbitrary Choice of $\alpha$: I also find that Appendix A does not sufficiently address the arbitrary selection of $\alpha$. Does the model’s performance change with varying $\alpha$, and would such variation impact the conclusions drawn?

* Performance and Budget Constraints: Finally, I feel that the average improvement achieved is often marginal. Moreover, the performance degradation at budget $b$ appears too severe to be practical in real-world settings. Could the authors provide insight on the specific budget value 1024 < $b < 6711$ perhaps (2048, 4096) at which both their approach and other key-value eviction strategies yield comparable performance to the baseline? Additionally, is there a discernible advantage to the proposed method under these conditions?

**Questions:**

Please see weaknesses.

---

> ### Author Response · Authors · 2024-11-22
> **Reply to Reviewer 19en | Part 1**
>
> Thank you for your valuable suggestions on our paper. Based on your feedback, we have revised the manuscript and addressed your concerns as follows:
>
> ## **Reply to some writing and formatting issues**
> > Section 3.2: In my opinion, this section needs to be reformatted and partially rewritten to be more concise with a clearer structure. As the most essential part of the paper, it is currently challenging to read, with many notations introduced ambiguously and references placed somewhat haphazardly.
>
> Thank you for your valuable feedback! We have rewritten this section to provide more precise statements, included relevant references, and simplified the notation to enhance clarity.
>
> > Equation Alias: The author should create an alias for $\frac{N \odot A}{\sum_n N_i A_i}$ to simplify and streamline the equations that reference this expression.
>
>
> Thank you for your suggestion! We recognize the necessity of simplifying notation in theoretical formulations. As a result, we have revised the statement of Theorem 1 to explicitly define $A' = \frac{N \odot A}{\sum_n N_i A_i}$. Additionally, we have rewritten the subsequent theorems and their proofs, ensuring a clear and rigorous presentation.
>
>
> > Assumption 1 and Theorem 3: I believe these should form their own subsection,
>
> Thank you for your suggestion! We have revised our paper by splitting the original Section 3.3 into three sections: 3.3, 3.4, and 3.5. The new Section 3.3 focuses on addressing the argument: "Are attention weights sufficient for identifying critical cache entries?" Section 3.4 is dedicated to presenting the core method, while Section 3.5 emphasizes its theoretical analysis.
>
>
>
> ## **Reply to unclear points**
>
> > Additionally, there are several points I still find unclear...
> >
> > These statements critique the empirical reliance on attention weights as indicators of importance. Could the authors clarify how they reconcile these criticisms with their own assumption about using attention scores to represent the importance of key-value entries?
>
> Thanks for your question! We do allocate a small portion of the budget in Assumption 1(Stage 1 in our algorithm) to collect some critical cache entries. However, this does not contradict our core argument that solely relying on attention weights is insufficient for identifying critical cache entries. The evidence lies in the fact that **the majority of our budget** is allocated to simultaneously consider **both attention weights and the norm of projected value states** for improved critical cache selection (Stage 2 in our algorithm). While it is feasible to rely solely on stage 2, omitting stage 1 (Assumption 1) entirely, our algorithm still demonstrates significant benefits as shown in **the ablation study(Appendix D)**. However, the inclusion of stage 1 provides a stronger theoretical foundation and consistently enhances performance, as demonstrated by our analysis and experimental results.
>
> Our algorithm operates in two distinct stages. Stage 1 (under Assumption 1) utilizes a small portion of the budget $b'$ to collect critical cache entries solely based on attention weights.
> Stage 2, which forms the core of our method, allocates the majority of budget $b''$ to account for both attention weights and the norm of projected value states. This two-stage algorithm is specifically designed to constrain the worst-case perturbations during selection.
>
> Overall, stage 1 in our algorithm serves as a preparatory step rather than the primary focus of our work. However, its inclusion is necessary for both theoretical and practical reasons.
> From a theoretical perspective, as shown in Theorem 3, the collection of $b'$ in stage 1 ensures the objective of stage 2 to align with constraining output perturbations, i.e. $\hat{\theta}$, enabling the effective selection of critical KV cache entries.
> From a practical standpoint, we demonstrate the necessity of the first stage in the ablation study presented in Appendix D. When the first stage is removed $\alpha = 0$, the algorithm's performance exhibits a noticeable decline.
> In this revised version of our paper, we have made the relationship and distinction between the two stages more explicit, ensuring a clearer presentation of their respective roles and contributions.

---

> ### Author Response · Authors · 2024-11-22
> **Reply to Reviewer 19en | Part 2**
>
> ## **Reply to arbitrary choice of $\alpha$**
>
> > I also find that Appendix A does not sufficiently address the arbitrary selection of α. Does the model’s performance change with varying α, and would such variation impact the conclusions drawn?
>
> This is a good question.
>
> First, we would like to clarify that even when $\alpha$ is set to 0, our algorithm remains effective. This is supported by the ablation study presented in Appendix D: although the performance is slightly degraded compared to the setting where $\alpha = 0.25$ (as used in the main experiments), it still significantly outperforms the vanilla baseline.
> Second, we agree with your observation that the model’s performance changes with varying $\alpha$. This relationship can be better understood through a conceptual lens: $\alpha$ represents the modification strength of our algorithm relative to conventional Top-K selection. When $\alpha = 0$, the algorithm fully incorporates both the norm of the projected value states and the attention weights in its decision-making. Conversely, when $\alpha = 1$, our algorithm degrades into a vanilla attention-weights-based Top-K selection (vanilla SnapKV Pyramid or AdaKV). Both extremes, as shown by our ablation and main experiments, are suboptimal choices.
>
> *Then why do we choose $\alpha = 0.25$ in this paper?*
>
> Given the computational overhead of hyperparameter tuning in LLMs on the LongBench evaluation, conventional grid search for $\alpha$ is impractical. Instead, we followed a theoretically guided approach: we determined that an $\alpha$ portion of 0.25 sufficiently satisfies our Assumption 1 across most attention heads. While it is true that a small number of heads may violate this assumption, their adverse impact is negligible, as the advantages gained from other heads overwhelmingly compensate for these exceptions. Further optimizing $\alpha$ on a finer granularity—for instance, setting $\alpha$ on a per-head basis to address such outliers—could offer additional benefits. However, this would require careful consideration of multiple factors, including model-specific characteristics, budget constraints, and downstream tasks, thereby introducing considerable algorithm complexity. Consequently, we think this is an interesting direction for future research.

---

> ### Author Response · Authors · 2024-11-22
> **Reply to Reviewer 19en | Part 3**
>
> ## **Reply to performance and budget constraints**
>
> > the average improvement achieved is often marginal.
>
> We clarify that the improvement achieved by our method is not marginal.
>
> 1. Significant Budget Savings: As demonstrated in Figure 1 of our paper, our method achieves an average additional budget saving of 19.7% and up to 34% compared to current SOTA cache eviction methods, all while maintaining the same quality score. This is primarily due to the inherent trade-off in cache eviction tasks, where significant budget investment typically yields only minimal accuracy improvements. Consequently, any enhancement in accuracy is highly valuable. This means our method allows for substantial additional budget savings while maintaining equivalent accuracy, directly translating into significant KV cache memory reductions and accelerating subsequent decoding processes.
>
> 2. Robust and Universal: Our algorithm is a robust and broadly applicable cache eviction enhancement method. It can seamlessly integrate with any current SOTA methods based on attention weights to further boost performance. Given its simplicity and generality, the observed performance improvements are quite remarkable.
>
>
>
> > the performance degradation at budget b appears too severe to be practical in real-world settings.
>
> Thanks for your question. We would like to clarify that, on the contrary, the accuracy degradation is minimal. For instance, as shown in Table 3, the Llama-3.1-8B model achieves an average quality score of 49.20 with the full KV cache and an average sequence length of 6711. With Ada-KV combined with our method, using a budget of 1024 (6.6x compression ratio) results in only a 2.2% reduction in the quality score (48.13). At a budget of 512 (13.2x compression ratio), the quality reduction is merely 3.07% (47.70). In practical applications, this means that with just a 3% accuracy trade-off, you can compress a sequence by 13.2x. Under the same KV cache capacity, this represents a highly practical and efficient trade-off for real-world scenarios.
>
> Notably, increasing compression from 6.6x to 13.2x costs only 0.87% in accuracy, underscoring the value of even slight accuracy improvements. Such gains can enable further compression, yielding significant practical benefits in real-world applications.
>
>
>
>
> > Could the authors provide insight on the specific budget value 1024 < b<6711 perhaps (2048, 4096) at which both their approach and other key-value eviction strategies yield comparable performance to the baseline? Additionally, is there a discernible advantage to the proposed method under these conditions?
>
>
>
> Thanks for your question! The table below provides the accuracy of various methods under a budget of 4096, along with their respective accuracy loss ratios compared to the Full Cache scenario.
> Indeed, when the budget is increased to a large value, such as 4096, the average compression ratio becomes minimal. Consequently, all methods—including SnapKV, Pyramid, and AdaKV, as well as their enhanced versions using our approach—achieve nearly identical lossless accuracy, around 0.6% loss on LongBench.
>
> However, we emphasize that practical applications prioritize smaller compression budgets, such as budget 128-1024, which are more relevant for real-world deployment. For example, using SnapKV with our enhancements, increasing the compression ratio from 1.6x to 6.6x (budget 4096 → 1024) results in only a modest accuracy drop of 1.28% (from 0.65% to 1.93%), demonstrating an appealing trade-off.
>
> It is also important to note that cache eviction is primarily useful in scenarios requiring high compression ratios, typically exceeding 10x. At such ratios, it demonstrates outstanding advantages over full cache methods, making it motivating for real-world applications. This is supported by related works, such as PyramidKV and SnapKV, which also focus on compression budgets ranging from 128 to 1024 in their LongBench evaluation.
>
>
>
>
>
>
>
> | Llama-3.1-8B      Full KV (49.20) | Snap w/o ours    | Snap w/ ours     | Pyramid w/o ours | Pyramid w/ ours  | Ada w/o ours     | Ada w/ ours      |
> |-----------------------------------|------------------|------------------|------------------|------------------|------------------|------------------|
> | Budget 4096 Ave.                  | **48.90(0.61%)** | 48.88(0.65%)     | 48.75(0.92%)     | **48.85(0.71%)** | **48.94(0.53%)** | 48.90(0.61%)     |
> | Budget 1024 Ave.                  | 48.08(2.28%)     | **48.25(1.93%)** | 47.92(2.60%)     | **48.09(2.26%)**  | 48.01(2.42%)     | **48.13(2.17%)** |
>
>
>
> We hope our responses have helped address your concerns. Should you have any additional questions, we would be more than happy to provide further clarification.

---

> > ### Comment · Reviewer_19en · 2024-11-23
> > **Reply to Author rebuttal.**
> >
> > I want to thank the authors for their detailed responses. I am satisfied with them and will raise my score.

---

> > > ### Author Response · Authors · 2024-11-25
> > > **Thanks for Your Timely Reply**
> > >
> > > We sincerely appreciate your response and recognition of our work. Your valuable suggestions have played a crucial role in enhancing the quality of our manuscript!

---

### Official Review · Reviewer_YE2P · 2024-11-03

**Soundness:** 3
**Presentation:** 3
**Contribution:** 3
**Rating:** 6
**Confidence:** 4

**Summary:**

This research examines KV cache compression by analyzing its impact on output perturbation. The study identifies critical KV cache elements by evaluating their maximum potential effect on output disruption, taking into account both value vector norms and attention score matrices. Through extensive testing across 16 datasets, the proposed methodology demonstrates consistent performance improvements.

**Strengths:**

1. KV cache compression represents a critical and increasingly relevant challenge in recent literature.
2. The study evaluates three distinct KV cache eviction strategies, each yielding comparable performance improvements.
3. Both the analytical framework and the implemented algorithm are reasonable.

**Weaknesses:**

This study can be further enhaned by enriching its experiments and empirical analyses (see suggestions & questions for more details).

**Questions:**

# Writing
On the proposed kv cache eviction algorithm, it would be better to discuss the connection and difference with FastGen [1] and MInference [2] in more details, given the similarity of the resulting algorithm. Also, as to the output perturbation, I feel it would be helpful to discuss the  existing success of this concept's adaptation in other deep learning algorithms (e.g., [3] and [4])

# Analyses
Additional theoretical analysis could strengthen the output perturbation perspective. A key area for investigation is the correlation between worst-case output perturbation predictions and actual perturbation measurements in real-world applications.

# Experiments
Intuitively, the proposed algorithm applies not only to long-context tasks, but to all generation tasks. However, in the evaluation, only long-context tasks are used for evaluation. I highly suggest the author to evaluate the proposed algorithm in other popular LLM benchmarks (e.g.,  AlpacaEval). This is necessary to understand the performance of the proposed algorithm on a wider spectrum of scenarios.

1. Model Tells You What to Discard: Adaptive KV Cache Compression for LLMs
2. MInference 1.0: Accelerating Pre-filling for Long-Context LLMs via Dynamic Sparse Attention
3. Understanding the Difficulty of Training Transformers
4. Catformer: Designing Stable Transformers via Sensitivity Analysis

---

> ### Author Response · Authors · 2024-11-22
> **Reply to Reviewer YE2P | Part 1**
>
> Thank you for your feedback and recognition of our work. We have revised our manuscript based on your suggestions, which has further improved its quality.
>
> ## **Reply to Question of Writing**
> > On the proposed kv cache eviction algorithm, it would be better to discuss the connection and difference with FastGen [1] and MInference [2] in more details, given the similarity of the resulting algorithm. Also, as to the output perturbation, I feel it would be helpful to discuss the existing success of this concept's adaptation in other deep learning algorithms (e.g., [3] and [4])
>
>
>
> Thanks for your suggestions! We have added an additional section in Appendix J discussing the relationship between these works with our method.
> Below is a summary of our key points:
>
>
> We have cited two related works, **FastGen** and **Minference**, and provided a detailed discussion of their relationship to our work. Notably, these two works stand out for their use of adaptive strategies, leveraging distinct critical cache selection mechanisms tailored to the characteristics of different attention heads. For instance, some heads employ attention weight-based selection, while others utilize fixed patterns, such as recent-window-based selection. In our paper, we demonstrate that our method effectively enhances the performance of solely attention-weight-based selection. This suggests a promising direction for future research: further enhancing strategies specifically designed for heads using attention-weight-based selection, thereby improving the overall quality of adaptive methods.
>
> We have also expanded the discussion on prior perturbation-based analyses, including **Catformer** and **Admin**, both of which represent valuable contributions to neural network interpretability.
> The **Catformer** applies perturbation analysis to design more stable architectures, while **Admin** examines output perturbations in residual blocks to improve training stability.  **Our work** extends perturbation analysis to the attention mechanism, focusing on perturbations caused by cache eviction. We propose improved critical cache selection metrics that mitigate these perturbations. These works collectively address perturbations in different components, such as residual connections and attention mechanisms. Future research could integrate these perturbation analysis strategies to conduct a more comprehensive investigation into network perturbations. Such an approach could guide the development of more robust architectures, training methodologies, and inference schemes.
>
>
> ## **Reply to Question of Analyses**
> > Additional theoretical analysis could strengthen the output perturbation perspective.
> > A key area for investigation is the correlation between worst-case output perturbation predictions and actual perturbation measurements in real-world applications.
>
> Thank you for highlighting this critical aspect. However, conducting a comprehensive theoretical analysis of actual perturbations remains challenging. The gap between theoretical perturbation and real-world perturbation measurements arises from inherently complex and difficult-to-analyze factors, such as the cumulative effects of multiple Transformer layers, residual connections, and feed-forward networks.
>
> Nonetheless, we attempted further analysis of output perturbations. As detailed in Appendix F, we demonstrate that the previously proposed method, based solely on attention weights, is equivalent to optimizing a relaxed upper bound, $\hat{\theta}^{relax}$. In contrast, our algorithm optimizes a tighter upper bound $\hat{\theta}$. While this does not guarantee that our approach will yield a strictly better solution, intuitively, an algorithm designed to optimize a tighter bound often achieves better results.
>
> From the perspective of actual perturbation measurements, in Section 5, we provide an empirical analysis under various conditions. As shown in Figure 2, our method significantly reduces perturbations across most heads in all tested models. Furthermore, in Figure 3, we demonstrate that these reductions propagate effectively, leading to a notable decrease in final-layer perturbations under various budget constraints. This, in turn, mitigates the quality degradation often observed with conventional cache eviction methods. To some extent, these results also validate the effectiveness of our theory. We hope this addresses your concerns.

---

> ### Author Response · Authors · 2024-11-22
> **Reply to Reviewer YE2P | Part 2**
>
> ## **Reply to Question of Experiments**
> >I highly suggest the author to evaluate the proposed algorithm in other popular LLM benchmarks (e.g., AlpacaEval). This is necessary to understand the performance of the proposed algorithm on a wider spectrum of scenarios.
>
> Thanks for your suggestion! Following your recommendation, we test the performance of SnapKV and SnapKV with ours using Llama-3.1-8B in AlpacaEval with Deepseek-chat and Yi-Large as Auto-annotators. The results, which demonstrate an impressive performance improvement with our method, have been included in Appendix L. Regarding compression settings, we analyze AlpacaEval and find that the average input length is 36.5 tokens, while the average generated length is 567 tokens. Thus, we conduct compression experiments by compressing the cache size to 64 tokens when the context length exceeds 256 tokens—approximately half of the average generated length. As shown in the table, our enhanced version of SnapKV significantly outperforms the original SnapKV version. For instance, using Yi-Large as the Auto-annotator, our method improves the scores for **Win Rate** and **LC Win Rate** from **19.61** and **18.76** to **21.20** and **19.79**, respectively.
>
> | Method                            | Win Rate | LC Win Rate |
> |-----------------------------------|----------|-------------|
> | Deepseek-chat Llama-3.1-8B        | 19.84    | 16.71       |
> | Deepseek-chat Llama-3.1-8B SnapKV    | 13.63    | 11.10       |
> | Deepseek-chat Llama-3.1-8B SnapKV w/ ours | **14.46**    | **12.05**       |
> | Yi-large Llama-3.1-8B             | 25.72    | 24.34       |
> | Yi-large Llama-3.1-8B SnapKV          | 19.61    | 18.76       |
> | Yi-large Llama-3.1-8B SnapKV w/ ours      | **21.20**   |**19.79**      |
>
> We hope our responses have helped address your concerns. Should you have any additional questions, we would be more than happy to provide further clarification.

---

> ### Author Response · Authors · 2024-11-25
> **Kindly Request**
>
> Dear Reviewer YE2P,
>
> We hope this message finds you well. Thank you for your thoughtful reviews of our submission.
>
> We have carefully addressed your concerns and made the necessary revisions to improve our work. As the discussion phase nears its conclusion, we kindly request your evaluation of our responses. If our revisions have satisfactorily addressed your concerns, we would be deeply grateful for your reconsideration of the overall score.
>
> Should you have any additional points that require clarification, please do not hesitate to reach out—we would be happy to engage in further discussion. Thank you once again for your valuable feedback and contributions to improving our work.
>
> Best regards,
>
> Authors.

---

> > ### Comment · Reviewer_YE2P · 2024-12-01
> > **Rebuttal Response**
> >
> > Thanks the author for the new results. These results addressed my concerns, and I am maintaining my score.
> >
> > On a side node, another suggestion is to include more analyses on the gap between full cache and various compared methods. This is because that, even after augmented by the proposed method, these is still a substantial gap on both alpacaleval and longbench. I believe the final goal of the kv eviction is to match the performance of the full cache model, while saving memories. If this goal is not attainable for the model of the same sizes, probably it is also worth comparing the performance of the proposed model (e.g., on qwen 2.5 7B) v.s. using full cache on a smaller model (e.g., on qwen 2.5 3B).

---

### Official Review · Reviewer_hVtB · 2024-11-04

**Soundness:** 3
**Presentation:** 3
**Contribution:** 3
**Rating:** 5
**Confidence:** 4

**Summary:**

This paper optimizes the critical KV selection for existing KV eviction methods during LLM inference. To identify critical KV, this paper uses L1 distance to measure the perturbation of attention outputs w./w.o. KV eviction under certain budgets, and draws the conclusion that both attention weights and value states projected through parameter matrix are important for reducing the output perturbation. It then proposes a perturbation-constrained selection algorithm which adopts top-k sampling to select two sets of KV: one for high attention weights and the other for both the norm of the projected value states and attention weights, respectively. Experimental results show that when integrating this selection algorithm into existing KV eviction methods like SnapKV, AdaKV, the output quality can be improved.

**Strengths:**

1.	Well-motivated: this paper points out that current KV eviction methods relying solely on attention weights, which is not a proper metric for identifying critical KV.
2.	Sufficient theoretical analysis: the conclusion aligns with the motivation.
3.	Good writing: easy to follow.

**Weaknesses:**

In general, the contributions are incremental, many claims lack supports and the experiments are not sufficient enough.

1.	Many claims lack further explanation or theoretical supports. For example, why a subset of KV is sufficient for LLM inference? Instead of claiming that it is a commonly accepted assumption from empirical studies, more insights from theoretical perspective are expected since this paper emphasizes its theoretical contributions.
(1) Why L1 distance is a good metric for measuring the output perturbation? L1 distance is well-known for its sparsity-sensitivity, but the attention outputs are dense. Any further explanation or theoretical supports for this metric?
(2) What’s the relationship between output perturbation and the generation quality? Perturbation measures the deviation from the original output, but does this deviation have a significant impact on determining the output token?
(3) Why set alpha to 0.25? The attention weights obey the power-law distribution, but why setting a fixed hyper-parameter is reasonable for any models and input distributions? Deriving this parameter directly from the attention weight distribution seems to be more reasonable.

2.	The comparison is far from sufficiency, omitting many important baselines. For example, StreamingLLM [1] should be considered since it is an important empirical work. This paper needs this comparison to verify the gains from theorical analysis. Besides, the naïve attention mechanism (without any KV dropping) should also be included. The authors should demonstrate the generation quality gap between eviction methods and the ground-truth.

[1] Xiao, Guangxuan, et al. "Efficient streaming language models with attention sinks." arXiv preprint arXiv:2309.17453 (2023).

**Questions:**

Justify claims. Add more experiments comparing with baselines.

---

> ### Author Response · Authors · 2024-11-22
> **Reply to Reviewer  hVtB | Part 1**
>
> Thank you for your effort and feedback; it means a lot to us. We will address your concerns below, and the related discussions have been added to the appendix of the revised manuscript to offer a broader perspective on our work.
> ## **Reply to Weakness 1**
>
> > For example, why a subset of KV is sufficient for LLM inference? Instead of claiming that it is a commonly accepted assumption from empirical studies, more insights from theoretical perspective are expected since this paper emphasizes its theoretical contributions.
>
> Thank you for your suggestions. We have included a detailed theoretical analysis in Appendix F for this assumption.  In Appendix F, we derive a relaxed upper bound $\hat{\theta}^{relax}$ for output perturbation that directly correlates with attention weights. Considering H2O's observation[1] that attention weights follow a power-law distribution—only a small fraction of KV cache entries have significant attention weights, while the majority exhibit weights close to zero. Thus discarding most KV cache entries with near-zero attention weights has minimal impact on this bound. Consequently, the perturbation to the actual output is also bounded by this upper bound. This result provides a theoretical analysis for the widely accepted assumption that a subset of KV entries suffices for LLM inference.
>
> [1] Zhenyu Zhang, Ying Sheng, Tianyi Zhou, Tianlong Chen, Lianmin Zheng, Ruisi Cai, Zhao Song,
> Yuandong Tian, Christopher R ́e, Clark Barrett, et al. H2o: Heavy-hitter oracle for efficient generative inference of large language models. Advances in Neural Information Processing Systems,
> 36, 2024b
>
>
>
> > (1)Why L1 distance is a good metric for measuring the output perturbation? L1 distance is well-known for its sparsity-sensitivity, but the attention outputs are dense. Any further explanation or theoretical supports for this metric?
>
>
>
> This is an interesting question, and we have included our views in Appendix G in the revised manuscript. In this paper, we employ the simplest distance metric, L1 distance, for our analysis, while future work could explore the use of L2 distance or other metrics. We chose $L_1$ distance for the following two reasons:
>
> 1. **Theoretical Perspective**: The L1 distance, as a straightforward metric, facilitates the construction and derivation of our theoretical framework. As you mentioned, $L_1$ distance is likely to be sparsity-sensitive. However, the $L_1$  distance used in this work could encourage the sparsity of perturbations rather than directly influencing the attention output itself. Therefore, we believe that the sparsity induced by $L_1$ regularization does not compromise the quality of the results.
>
> 2. **Practical Perspective**: Considering that BF16 (half-precision floating-point) is commonly used in inference computations, the L1 distance operations derived from our algorithm provide better numerical precision. In contrast, metrics such as $L_2$ distance may introduce numerical precision issues due to the squaring and square-rooting operations inherent in their computation.
>
> Fundamentally, the choice of distance metric is orthogonal to our primary contributions. We also appreciate your suggestions and plan to further investigate the impact of different distance metrics in future work.

---

> ### Author Response · Authors · 2024-11-22
> **Reply to Reviewer hVtB | Part 2**
>
> ## **Reply to Weakness 1(Continued)**
> > (2)What's the relationship between output perturbation and the generation quality? Perturbation measures the deviation from the original output, but does this deviation have a significant impact on determining the output token?
>
> Thanks for your question! Output perturbation indeed has a significant impact on determining the output token thereby affecting generation quality. We directly measure the difference in token vocabulary distributions between SnapKV (and SnapKV with our method) and the original full KV Cache case on 200 samples from the Qasper dataset (QA dataset in Longbench). The comparison, shown in the table below, focuses on the first four generated tokens. Regardless of the token position or whether $L_1$ distance or KV divergence is used as the metric, our method **consistently reduces perturbation in the token vocabulary distribution** for most samples. This demonstrates that our enhanced approach significantly improves consistency with the token distribution of the uncompressed case, effectively mitigating the quality degradation caused by compression.
>
> The number of samples with lower perturbation in token vocabulary distributions for generation (higher is better).
>
> | L1 distance   | 1st  token   | 2nd   token  | 3rd  token   | 4th  token   |
> |----------------|---------|---------|---------|---------|
> | SnapKV         | 71      | 72      | 68      | 69      |
> | SnapKV w/ Ours | **129** | **128** | **132** | **131** |
>
>
>
> | KL divergence  | 1st  token   | 2nd   token  | 3rd   token  | 4th   token  |
> |----------------|---------|---------|---------|---------|
> | SnapKV         | 73      | 83      | 77      | 65      |
> | SnapKV w/ Ours | **127** | **117** | **123** | **135** |
>
>
> Next, we approach this from a different perspective and incorporate this discussion into Appendix I in the revised manuscript.
>
> In LLM computations, the attention output serves as an input to the FeedForward Neural Network (FFN) module, producing outputs that are subsequently passed to the Language Model (LM) head to generate the token vocabulary distribution. Our algorithm specifically aims to reduce perturbations in attention outputs caused by cache eviction methods. This reduction lowers the perturbations in the inputs to downstream network components (FFN and LM head), thereby mitigating perturbations in the final token vocabulary distribution. Consequently, our method reduces the impact of cache eviction on output token generation.
>
> This conclusion is supported both theoretically and empirically. Theoretically, the relationship between reduced input perturbations and diminished output variations is well-established, as seen in Lipschitz continuity [2-3], which posits that functions with bounded Lipschitz constants exhibit smaller output differences for smaller input differences. This principle is consistent with our approach to minimizing attention output perturbations, thereby ensuring subsequent generated tokens. Similarly, FFN pruning techniques in LLMs[4-5] also demonstrate the practical success of minimizing perturbations to downstream layers' outputs, further validating our strategy.
>
> [2] Harry Dong, Beidi Chen, and Yuejie Chi. Prompt-prompted adaptive structured pruning for efficient llm generation. 2024.
>
> [3] Mingjie Sun, Zhuang Liu, Anna Bair, and J. Zico Kolter. A simple and effective pruning approach for large language models, 2024.
>
> [4] Yuesheng Xu and Haizhang Zhang. Uniform convergence of deep neural networks with lipschitz continuous activation functions and variable widths. IEEE Transactions on Information Theory,70(10):7125–7142, 2024.
>
> [5] Grigory Khromov and Sidak Pal Singh. Some fundamental aspects about lipschitz continuity of neural networks, 2024.

---

> ### Author Response · Authors · 2024-11-22
> **Reply to Reviewer hVtB | Part 3**
>
> ## **Reply to Weakness 1(Continued)**
>
> > Why set alpha to 0.25? The attention weights obey the power-law distribution, but why setting a fixed hyper-parameter is reasonable for any models and input distributions? Deriving this parameter directly from the attention weight distribution seems to be more reasonable.
>
> Setting $\alpha$ as an adaptive threshold for individual heads can indeed improve quality further. However, after careful consideration, we decided not to adopt this approach for the following reasons:
>
> 1. **Increased Complexity and limiting practical application**: To implement an adaptive threshold, distinct values of $\alpha$ must be set for each head in different LLMs, across various budget configurations, and even for different input tasks. In the case of a 7B or 8B model, this requires setting unique $\alpha$ values for 32 $\times$ 32 heads. Whether through offline or online analysis, this approach introduces substantial search overhead, which further complicates the process and restricts its practical applicability.
>
> 2. **A Simple, Effective Solution with a Fixed $\alpha$ = 0.25**: A fixed $\alpha = 0.25$ serves as a simple and robust solution. Visualizations in Appendix A demonstrate that this fixed threshold satisfies Assumption 1 for the majority of heads, leaving a 75% budget to execute our novel selection strategy. While there are exceptions in a few heads in shallow layers, our analysis shows that these discrepancies are effectively mitigated by the advantages of subsequent layers. By the final Transformer layer, output perturbations are significantly reduced.
>
> Overall, a straightforward and robust $\alpha$ = 0.25 setting allows our algorithm to achieve up to 34% cache budget savings, as illustrated in Figure 1, under comparable generation quality across three existing eviction algorithms (SnapKV, Pyramid, and AdaKV) tested on two LLMs.

---

> ### Author Response · Authors · 2024-11-22
> **Reply to Reviewer hVtB | Part 4**
>
> ## **Reply to Weakness 2**
>
> > For example, StreamingLLM should be considered since it is an important empirical work.
>
> Based on your suggestion, we have added StreamingLLM as a baseline for comparison in Appendix H.
>
> Since StreamingLLM does not rely on attention weights based selection for cache eviction and instead retains only the several initial cache entries and those within the recent window, it can't be combined with our algorithm. Due to the non-selective nature of its eviction strategy, StreamingLLM performs significantly worse than existing selection based cache eviction methods, e.g. SnapKV.
>
> For instance, consider SnapKV w/ ours on Llama3.1-8B as a representative example of selection-based cache eviction approaches. The quality scores of StreamingLLM under cache budgets ranging from 128 to 1024 are 38.42, 39.75, 41.52, and 42.56, respectively. In contrast, the selection-based cache eviction methods achieve significantly higher scores of 43.25, 45.66, 47.41, and 48.25 under the same budget configurations. Detailed results can be found in Appendix H.
>
>
>
> | Budget                     | 128       | 256       | 512       | 1024      | Full Cache |
> |----------------------------|-----------|-----------|-----------|-----------|------------|
> | Llama3.1-8B StreamingLLM   | 38.42     | 39.75     | 41.52     | 42.56     | 49.20      |
> | Llama3.1-8B SnapKV w/ ours | **43.25** | **45.66** | **47.41** | **48.25** | 49.20      |
>
> > Besides, the naïve attention mechanism (without any KV dropping) should also be included. The authors should demonstrate the generation quality gap between eviction methods and the ground-truth.
>
>
>
> The naïve attention mechanism without KV dropping is described in Tables 1, 2, and 3 of the paper, labeled as "Full Cache", where its performance is thoroughly presented. Using the SOTA method AdaKV w/ours as an example, when the existing cache eviction algorithm is enhanced by our method, the quality loss is only 1.45% at a 1024 budget with compression. For a further 8x compression at a 128 budget, the quality loss only increases to 9.2%.
>
>
>
>
> | 1024 Budgets | Naive attention(Full Cache) | SnapKV w/o Ours | SnapKV w/ Ours   | AdaKV w/o Ours | AdaKV w/ Ours    |
> |--------------|-------------------------------|-----------------|------------------|----------------|------------------|
> | Llama-3.1-8B | 49.2                          | 48.08(2.28%)    | **48.25(1.93%)** | 48.01(2.42%)   | **48.13(2.17%)** |
> | Mistral-7B   | 47.72                         | 46.71(2.12%)    | **46.84(1.85%)** | 46.89(1.74%)   | **47.03(1.45%)** |
>
>
> | 128 Budgets  | Naive attention(Full Cache) | SnapKV w/o Ours | SnapKV w/ Ours    | AdaKV w/o Ours | AdaKV w/ Ours    |
> |--------------|-------------------------------|-----------------|-------------------|----------------|------------------|
> | Llama-3.1-8B | 49.2                          | 42.70(13.21%)   | **43.25(12.09%)** | 43.94(10.7%)   | **44.26(9.04%)** |
> | Mistral-7B   | 47.72                         | 41.12(13.83%)   | **41.69(12.64%)** | 42.53(10.9%)   | **43.34(9.2%)**  |
>
>
>
>
> ## **Additional long-text retrieval Benchmark ：Needle-in-the-Haystack Test**
>
> We have also added a widely used benchmark for long-text retrieval task, the needle-in-the-haystack test, with detailed results provided in Appendix E. Using the Llama-3.1-8B model and a budget of 128, we extended inference to the model's maximum supported length of 128K, achieving up to a 1000x compression ratio. Under the Full Cache setting, the retrieval score reaches 87.58. Our algorithm improves the long-context retrieval performance of all cache eviction methods - SnapKV, Pyramid, and AdaKV - boosting their retrieval scores from 81.01, 82.53, and 82.71 to 82.01, 82.95, and 83.35, respectively. This further underscores the effectiveness of our algorithm in cache compression for long-sequence retrieval tasks.
>
> | 128 Budgets | Full Cache | w/o Ours | w/ Ours   |
> |-------------|------------|----------|-----------|
> | SnapKV      | 87.58      | 81.01    | **82.01** |
> | Pyramid     | 87.58      | 82.53    | **82.95** |
> | AdaKV       | 87.58      | 82.71    | **83.35** |
>
>
> We hope our responses have helped address your concerns. Should you have any additional questions, we would be more than happy to provide further clarification.

---

> ### Author Response · Authors · 2024-11-25
> **Kindly Request**
>
> Dear Reviewer hVtB,
>
> We hope this message finds you well. Thank you for your thoughtful reviews of our submission.
>
> We have carefully addressed your concerns and made the necessary revisions to improve our work. As the discussion phase nears its conclusion, we kindly request your evaluation of our responses. If our revisions have satisfactorily addressed your concerns, we would be deeply grateful for your reconsideration of the overall score.
>
> Should you have any additional points that require clarification, please do not hesitate to reach out—we would be happy to engage in further discussion. Thank you once again for your valuable feedback and contributions to improving our work.
>
> Best regards,
>
> Authors.

---

### Author Response · Authors · 2024-11-22
**Global Response**

We deeply appreciate all reviewers' careful evaluation and insightful feedback on our work. In response, we have revised our manuscript during the rebuttal phase, especially enhancing the clarity of Section 3. Furthermore, we included additional evaluation results and discussions in the appendix.
The revised version has been uploaded. If there are any additional concerns, please let us know, and we would be happy to engage in further discussions.
The key revisions are summarized as follows:

1. Presentation and Theoretical Improvements
    * **Section 3**: Reorganized for clearer presentation and more rigorous theoretical proofs.
    *  **Appendix F**: Added a theoretical analysis of previous methods from the perspective of output perturbation.
2. Experimental Additions
   * **Appendix E**: Introduced results from the **Needle-in-a-Haystack** Test.
   *  **Appendix K**: Added a new instruction-following benchmark, **AlpacaEval**, to demonstrate the broader applicability of our method.
   *  **Appendix H**:  Conducted a detailed comparison with an earlier empirical study (**StreamingLLM**).
3. Broader Discussion
   * **Appendix I**: Discussed the relationship between output perturbations and final generation quality.
   *  **Appendix J**: Expanded the related work discussion and future research directions.
   * **Appendix G**:  Discussed the choice of distance metric.

We hope these revisions effectively address the reviewers' concerns and contribute to enhancing the overall quality of the paper. If there are any remaining concerns, we would be happy to engage in further discussions. We look forward to hearing from you.

---

### Author Response · Authors · 2024-11-29
**General Response to Reviewers and ACs**

We would like to express our sincere gratitude to all the reviewers **[Reviewer 1 (hVtB), Reviewer 2 (YE2P), Reviewer 3 (19en), and Reviewer 4 (nKeS)]** for their thoughtful and supportive feedback. We are pleased that the reviewers recognized the following key aspects of our work:

* **Well-motivated (Reasonable)** work for identifying critical KV cache from a perturbation perspective. **[Reviewers 1, 2, 3, 4]**
* **Novel algorithm** for constraining worst-case perturbation through a two-stage selection process. **[Reviewers 3, 4]**
* **Solid theoretical foundation:** Our algorithm aligns well with the theoretical analysis. **[Reviewers 1, 2, 3, 4]**
* **Broad applicability** in enhancing existing methods. **[Reviewers 1, 2, 3, 4]**

In response to the reviewers' suggestions, we have refined the presentation of our paper and incorporated additional experiments. If there are any remaining concerns, we would be happy to discuss them further and look forward to your response.

---

### Meta-Review · Area_Chair_4EBC · 2024-12-21

**Metareview:**

This paper investigates the identification of critical key-value (KV) cache entries in large language model (LLM) inference through a novel perturbation-constrained selection algorithm. The problem of efficient KV cache compression is well-motivated, and the authors propose an innovative approach that combines attention weights and value norms to constrain output perturbation. The paper presents solid theoretical foundations and demonstrates the method’s ability to integrate with existing state-of-the-art (SOTA) cache eviction algorithms, achieving additional memory savings while maintaining output quality.

However, the paper has some limitations that prevent its acceptance at this stage. While the theoretical analysis is sound, several assumptions, such as the use of L1 distance and specific hyperparameter choices, lack comprehensive justification. Empirical evaluations, although extensive, are primarily focused on smaller models and long-context tasks, leaving broader applicability underexplored. Additionally, key baselines like larger LLMs and StreamingLLM are not fully addressed, which limits the understanding of the proposed method’s comparative advantages. Some parts of the paper, particularly the mathematical notations and key derivations, remain challenging to follow, which detracts from the clarity of the contributions.

It is unfortunate that I have to recommend rejecting this paper. The topic and approach are promising, but further refinement in theoretical justification, broader experimental evaluations, and improved clarity are necessary for the work to reach its full potential.

**Additional Comments On Reviewer Discussion:**

The authors have addressed several reviewer concerns constructively, including improving clarity in certain sections and adding new benchmarks. However, foundational concerns about theoretical assumptions and experimental breadth remain insufficiently addressed. While the revisions are appreciated, more substantial efforts are needed to enhance the robustness and accessibility of the paper.

---

### Decision · Program_Chairs · 2025-01-22

Reject